# Gestational age at birth and body size from infancy through adolescence: An individual participant data meta-analysis on 253,810 singletons in 16 birth cohort studies

Johan L. Vinther[1]*, Tim Cadman[2], Demetris Avraam[3], Claus T. Ekstrøm[4], Thorkild I. A. Sørensen[1,5], Ahmed Elhakeem[2], Ana C. Santos[6,7], Angela Pinot de Moira[1], Barbara Heude[8], Carmen Iñiguez[9,10,11], Costanza Pizzi[12], Elinor Simons[13,14], Ellis Voerman[15,16], Eva Corpeleijn[17], Faryal Zariouh[18], Gilian Santorelli[19], Hazel M. Inskip[20,21], Henrique Barros[6,7], Jennie Carson[22,23], Jennifer R. Harris[24], Johanna L. Nader[25], Justiina Ronkainen[26], Katrine Strandberg-Larsen[1], Loreto Santa-Marina[10,27,28], Lucinda Calas[8], Luise Cederkvist[1], Maja Popovic[12], Marie-Aline Charles[18], Marieke Welten[15,16], Martine Vrijheid[10,29,30], Meghan Azad[13,31,32], Padmaja Subbarao[33,34,35], Paul Burton[3], Puishkumar J. Mandhane[36], Rae-Chi Huang[22,37], Rebecca C. Wilson[38], Sido Haakma[39], Sílvia Fernández-Barrés[29,30,40], Stuart Turvey[41], Susana Santos[15,16], Suzanne C. Tough[42], Sylvain Sebert[26], Theo J. Moraes[33], Theodosia Salika[21], Vincent W. V. Jaddoe[15,16], Deborah A. Lawlor[2,43], Anne-Marie Nybo Andersen[1]

1 Section of Epidemiology, Department of Public Health, University of Copenhagen, Copenhagen, Denmark, 2 Population Health Science, Bristol Medical School, Bristol, United Kingdom, 3 Population Health Sciences Institute, Newcastle University, Newcastle, United Kingdom, 4 Section of Biostatistics, Department of Public Health, University of Copenhagen, Copenhagen, Denmark, 5 Novo Nordisk Foundation Center for Basic Metabolic Research, Faculty of Health and Medical Sciences, University of Copenhagen, Copenhagen, Denmark, 6 EPIUnit–Instituto de Saúde Pública, Universidade do Porto, Porto, Portugal, 7 Departamento de Ciências da Saúde Pública e Forenses e Educação Médica, Faculdade de Medicina, Universidade do Porto, Porto, Portugal, 8 Université Paris Cité and Université Sorbonne Paris Nord, Inserm, INRAE, Center for Research in Epidemiology and StatisticS (CRESS), Paris, France, 9 Department of Statistics and Operational Research, Universitat de València, València, Spain, 10 Spanish Consortium for Research on Epidemiology and Public Health (CIBERESP), Madrid, Spain, 11 FISABIO—Universitat Jaume I—Universitat de València Epidemiology and Environmental Health Joint Research Unit, València, Spain, 12 Cancer Epidemiology Unit, Department of Medical Sciences, University of Turin, Turin, Italy, 13 Section of Allergy and Immunology, Department of Pediatrics and Child Health, University of Manitoba, Winnipeg, Canada, 14 The Children's Hospital Research Institute of Manitoba (CHRIM), Winnipeg, Canada, 15 The Generation R Study Group, Erasmus MC, University Medical Center Rotterdam, Rotterdam, the Netherlands, 16 Department of Pediatrics, Erasmus MC–Sophia Children's Hospital, University Medical Center Rotterdam, Rotterdam, the Netherlands, 17 Department of Epidemiology, University of Groningen, University Medical Center Groningen, Groningen, the Netherlands, 18 Ined, Inserm, EFS, joint unit Elfe, Aubervilliers Cedex, France, 19 Born In Bradford, Bradford Institute for Health Research, Bradford, United Kingdom, 20 MRC Lifecourse Epidemiology Centre, University of Southampton, Southampton General Hospital, Southampton, United Kingdom, 21 NIHR Southampton Biomedical Research Centre, University of Southampton and University Hospital Southampton NHS Foundation Trust, Southampton, United Kingdom, 22 Telethon Kids Institute, Perth, Australia, 23 University of Western Australia, School of Population and Global Health, Perth, Australia, 24 Center for Fertility and Health, The Norwegian Institute of Public Health, Oslo, Norway, 25 Department of Genetics and Bioinformatics, Division of Health Data and Digitalisation, Norwegian Institute of Public Health, Oslo, Norway, 26 Center for Life-course Health research, University of Oulu, Oulu, Finland, 27 Biodonostia Health Research Institute, San Sebastian, Spain, 28 Health Department of Basque Government, Subdirectorate of Public Health of Gipuzkoa, San Sebastian, Spain, 29 ISGlobal, Barcelona, Spain, 30 Universitat Pompeu Fabra, Barcelona, Spain, 31 Developmental Origins of Chronic Diseases in Children Network (DEVOTION), Children's Hospital, Winnipeg, Canada, 32 Department of Food and Human Nutritional Sciences, University of Manitoba, Winnipeg, Canada, 33 Translational Medicine Program, Department of Pediatrics, The Hospital for Sick Children, Toronto, Canada, 34 Department of Medicine, McMaster University, Hamilton, Canada, 35 Dalla Lana School of Public Health, University of Toronto,

**Data Availability Statement:** The data used for this study is third-party data, and without legally permission of distribution or public sharing. Information about data access and governance for this study is explained in detail in a peer-reviewed

scientific paper by Prof. Vincent Jaddoe et al (2020): "The LifeCycle Project-EU Child Cohort Network: a federated analysis infrastructure and harmonized data of more than 250,000 children and parents". The principal investigators or home institutions administer permission to the data for external researchers: hence, access to the data is conditional on reasonable request and with approval by each cohort. A description of the data set and third-party sources are listed in Supplementary Materials, and displayed online at the EU Child Cohort Variable Catalogue (https://data-catalogue.molgeniscloud.org/catalogue/catalogue/#/) and the Maelstrom Catalogue (https://www.maelstrom-research.org/page/catalogue). Please, for data request find below cohort-specific contact details (email address and/or web). ALSPAC, United Kingdom Email: alspac-data@bristol.ac.uk Web: http://www.bristol.ac.uk/alspac/researchers/access/ AOF, Canada Email: stough@ucalgary.ca Web: https://allourfamiliesstudy.com/data-access/, http://allourfamiliesstudy.com/wp-content/uploads/2017/03/AOF-Access-and-Acknowledgement-Guidelines-March-2017-Sec.pdf BiB, United Kingdom Email: borninbradford@bthft.nhs.uk Web: https://borninbradford.nhs.uk/research/how-to-access-data/ CHILD, Canada Email: child@mcmaster.ca Web: https://childstudy.ca/for-researchers/data-access/ DNBC, Denmark Email: amna@sund.ku.dk Web: https://www.dnbc.dk/access-to-dnbc-data EDEN, France Email: etude.eden@inserm.fr ELFE, France Email: contact@elfe-france.fr Web: https://www.elfe-france.fr/en/the-research/access-to-data-and-questionnaires/ G21, Portugal Email: info@geracao21.com, catia.ferreira@ispup.up.pt GECKO, The Netherlands Email: e.corpeleijn@umcg.nl Web: https://www.umcg.nl/-/medisch-wetenschappelijk-onderzoek/gecko GEN R, The Netherlands Email: v.jaddoe@erasmusmc.nl Web: https://generationr.nl/researchers/collaboration/ INMA, Spain Web: https://www.proyectoinma.org/ MoBa, Norway Email: jennifer.harris@fhi.no Web: http://www.fhi.no/moba NFBC1986, Finland Email: sylvain.sebert@oulu.fi Web: https://www.oulu.fi/en/university/faculties-and-units/faculty-medicine/northern-finland-birth-cohorts-and-arctic-biobank/nfbc-aineistopyynto NINFEA, Italy Email: info@progettoninfea.it, lorenzo.richiardi@unito.it Web: https://www.progettoninfea.it/index_en SWS, United Kingdom Email: sws@soton.ac.uk Web: https://www.mrc.soton.ac.uk/sws/ RAINE, Australia Email: raineadmin-sph@uwa.edu.au Web: https://rainestudy.org.au/information-for-researchers/available-data/.

Toronto, Canada, **36** Department of Pediatrics, University of Alberta, Edmonton, Canada, **37** Edith Cowan University, School of Medicine and Health Sciences, Joondalup, Australia, **38** Institute of Population Health, University of Liverpool, Liverpool, United Kingdom, **39** University of Groningen, University Medical Center Groningen, Genomics Coordination Center, Groningen, the Netherlands, **40** CIBER Epidemiología y Salud Pública (CIBERESP), Madrid, Spain, **41** Department of Pediatrics, BC Children's Hospital, University of British Columbia, Vancouver, Canada, **42** Department of Community Health Sciences, Cumming School of Medicine, University of Calgary, Calgary, Canada, **43** MRC Integrative Epidemiology Unit at the University of Bristol, Bristol, United Kingdom

\* johan.vinther@sund.ku.dk

# Abstract

## Background

Preterm birth is the leading cause of perinatal morbidity and mortality and is associated with adverse developmental and long-term health outcomes, including several cardiometabolic risk factors and outcomes. However, evidence about the association of preterm birth with later body size derives mainly from studies using birth weight as a proxy of prematurity rather than an actual length of gestation. We investigated the association of gestational age (GA) at birth with body size from infancy through adolescence.

## Methods and findings

We conducted a two-stage individual participant data (IPD) meta-analysis using data from 253,810 mother–child dyads from 16 general population-based cohort studies in Europe (Denmark, Finland, France, Italy, Norway, Portugal, Spain, the Netherlands, United Kingdom), North America (Canada), and Australasia (Australia) to estimate the association of GA with body mass index (BMI) and overweight (including obesity) adjusted for the following maternal characteristics as potential confounders: education, height, prepregnancy BMI, ethnic background, parity, smoking during pregnancy, age at child's birth, gestational diabetes and hypertension, and preeclampsia.

Pregnancy and birth cohort studies from the LifeCycle and the EUCAN-Connect projects were invited and were eligible for inclusion if they had information on GA and minimum one measurement of BMI between infancy and adolescence. Using a federated analytical tool (DataSHIELD), we fitted linear and logistic regression models in each cohort separately with a complete-case approach and combined the regression estimates and standard errors through random-effects study-level meta-analysis providing an overall effect estimate at early infancy (>0.0 to 0.5 years), late infancy (>0.5 to 2.0 years), early childhood (>2.0 to 5.0 years), mid-childhood (>5.0 to 9.0 years), late childhood (>9.0 to 14.0 years), and adolescence (>14.0 to 19.0 years).

GA was positively associated with BMI in the first decade of life, with the greatest increase in mean BMI z-score during early infancy (0.02, 95% confidence interval (CI): 0.00; 0.05, $p < 0.05$) per week of increase in GA, while in adolescence, preterm individuals reached similar levels of BMI (0.00, 95% CI: −0.01; 0.01, $p$ 0.9) as term counterparts. The association between GA and overweight revealed a similar pattern of association with an increase in odds ratio (OR) of overweight from late infancy through mid-childhood (OR 1.01 to 1.02) per week increase in GA. By adolescence, however, GA was slightly negatively

**Funding:** This collaborative project received funding from the European Union's Horizon 2020 research and innovation programme (Grant Agreement No. 733206 LifeCycle, Grand Recipient VWVJ; Grant Agreement No. 824989 EUCAN-Connect, Grand Recipient AMNA). Please, see S1 Appendix for list of cohort-specific funding/support. DAL is supported by the UK Medical Research Council (MC_UU_00011/6) and British Heart Foundation (CH/F/20/90003 and AA/18/7/34219). RCW is supported by UKRI Innovation Fellowship with Health Data Research UK [MR/S003959/1]. The funders had no role in study design, data collection and analysis, decision to publish, or preparation of the manuscript.

**Competing interests:** I have read the journal's policy and the authors of this manuscript have the following competing interests: DAL has received support from Roche Diagnostics and Medtronic in relation to biomarker research that is not related to the research presented in this paper. The other authors have declared that no competing interests exist.

**Abbreviations:** BMI, body mass index; CI, confidence interval; DOHaD, developmental origins of health and disease; GA, gestational age; IPD, individual participant data; OR, odds ratio; WHO, World Health Organization.

associated with the risk of overweight (OR 0.98 [95% CI: 0.97; 1.00], $p$ 0.1) per week of increase in GA. Although based on only four cohorts ($n = 32,089$) that reached the age of adolescence, data suggest that individuals born very preterm may be at increased odds of overweight (OR 1.46 [95% CI: 1.03; 2.08], $p < 0.05$) compared with term counterparts.

Findings were consistent across cohorts and sensitivity analyses despite considerable heterogeneity in cohort characteristics. However, residual confounding may be a limitation in this study, while findings may be less generalisable to settings in low- and middle-income countries.

## Conclusions

This study based on data from infancy through adolescence from 16 cohort studies found that GA may be important for body size in infancy, but the strength of association attenuates consistently with age. By adolescence, preterm individuals have on average a similar mean BMI to peers born at term.

## Author summary

### Why was this study done?

- Conditions and exposures in early life is suggested to play an important role in development of cardiometabolic health outcomes, including body size.

- The majority of previous research focused on the impact of size at birth (i.e., birth weight), rather than the timing of birth (i.e., gestational duration).

- Moreover, we know less about how different contextual factors influence associations between early life risk factors and later body size.

### What did the researchers do and find?

- Our aim was to examine the association of gestational age with body mass index (BMI) and overweight from infancy through adolescence.

- We used data from 16 cohort studies in Europe, North America, and Australasia, including information on 253,810 mother–child dyads.

- We found that infants born preterm (<37 completed weeks of gestation) have a lower BMI and lower risk of overweight in infancy than their term counterparts and that this difference attenuates with age.

- In adolescence, BMI was similar between preterm and term peers, while there was an indication of an increased risk of overweight in very preterm individuals.

### What do these findings mean?

- Our study suggests that, although being born early, preterm infants on average reach the body size as their term peers before adulthood.

- In line with earlier findings, children born very preterm may even be at increased risk of overweight in adulthood, here already indicated at entrance to adulthood.

- This last finding must be interpreted with caution, as only four cohorts ($n$ = 32,089) contributed with data in adolescence.

- In addition, our study is based on data from high-income countries; hence, the findings are not generalisable to low- and middle-income country settings.

## Introduction

Today, one in ten infants are born preterm (<37 completed weeks' gestation) with an increased risk of perinatal mortality and morbidity that may persist and develop over the life-course [1,2,3]. Global estimates show an increase in preterm birth between 2000 and 2014, but the proportions vary between countries [4].

Previous systematic reviews and meta-analyses [5,6,7] have reported an association of gestational age (GA) at birth with conventional cardiometabolic risk factors and outcomes, including increased blood pressure, impaired glucose regulation, and insulin resistance in those born preterm [8,9,10,11]. An infant born preterm adapts to extrauterine conditions entering a phase of growth that possibly expresses a mismatch with the environment outside utero leading to alterations in body composition [12,13,14,15,16,17]. It has been hypothesised that these changes increase susceptibility to being overweight for preterm birth through various pathways and mechanisms, including catch-up weight [16,18,19,20,21,22]. However, later body size in preterm cohorts is not well characterised, and most studies define populations by birth weight rather than actual length of gestation [17,23]. It is recognised that determinants and consequences of gestational duration are quite different from those of foetal growth [23] and that birth weight reflects both gestational duration and foetal growth [24], hence being a potential intermediate variable on the causal pathway [25].

Studies have shown that infants born extremely (23 to 27 weeks gestation) and very preterm (28 to 31 weeks gestation) typically experience postnatal growth failure followed by catch-up weight and length gain within the first two years of life [20]. Growth in preterm children remains different from that of full term peers through childhood and into school age [26,27,28,29,30,31,32]. However, studies on growth in preterm cohorts across key stages of growth development [33] and at more advanced GA are scarce [10,20,34]. Several methodological considerations and sample characteristics complicate the interpretation and comparability of findings on the relationship between GA with later body size [5,6,35,36,37]. These include differences in study design; using birth weight as a proxy for GA; sample size; age at outcome; conditions under which variables are examined; type of statistical analysis; and availability of confounders.

In this study, we use the novel approach and unique opportunity of federated analysis of individual participant data (IPD) in a secure manner provided by the EU Child Cohort Network [38,39], an international network of European and Australasian birth cohort data. We base our study on 16 cohorts and 253,810 mother–child dyads, which enables us to extend previous research by including information on repeated body size measures during a long follow-up across a wide range of GA, and overcome the methodological limitations identified above.

The overall aim of this study was to determine the association between GA (completed weeks and clinical categories) and, respectively, body mass index (BMI) and overweight (including obesity) from infancy through adolescence in birth cohort studies representing diverse contexts.

## Methods

### Inclusion criteria and participating cohorts

In December 2019, we invited pregnancy and birth cohort studies within the EU Child Cohort Network from the LifeCycle and cohorts from the EUCAN-Connect consortia [38,39,40]. Cohorts were eligible for inclusion if they had information for live-born singletons on GA and at least one offspring measurement of BMI in one of six age-periods: early infancy (>0.0 to 0.5 years), late infancy (>0.5 to 2.0 years), early childhood (>2.0 to 5.0 years), mid-childhood (>5.0 to 9.0 years), late childhood (>9.0 to 14.0 years), and adolescence (>14.0 to 19.0 years).

The following 16 cohorts participated in the study: Avon Longitudinal Study of Parents and Children, UK (*ALSPAC*) (*n* = 10,452) [41]; All Our Families, Canada (*AOF*) (*n* = 2,263) [42]; Born in Bradford, UK (*BiB*) (*n* = 13,097) [43]; CHILD Cohort Study, Canada (*CHILD*) (*n* = 2,984) [44]; Danish National Birth Cohort, Denmark (*DNBC*) (*n* = 81,117) [45]; The EDEN mother–child cohort on the prenatal and postnatal determinants of child health and development, France (*EDEN*) (*n* = 1,765) [46]; French Longitudinal Study of Children, France (*ELFE*) (*n* = 15,506) [47]; The Generation 21 Birth Cohort, Portugal (*G21*) (*n* = 6,439) [48]; The GECKO Drenthe Cohort, the Netherlands (*GECKO*) (*n* = 2,768) [49]; The Generation R Study, the Netherlands (*GEN R*) (*n* = 8,641) [50]; The Environment and Childhood Project, Spain (*INMA*) (*n* = 1,936) [51]; The Norwegian Mother, Father and Child Study, Norway (*MoBa*) (*n* = 86,553) [52]; The Northern Finland Birth Cohort 1986, Finland (*NFBC1986*) (*n* = 8,325) [53,54,55]; The NINFEA (Nasita e INFanzia: gli Effetti dell'Ambiente) birth cohort study, Italy (*NINFEA*) (*n* = 6,515) [56]; The Raine Study, Australia (*The Raine Study*) (*n* = 2,443) [57]; and The Southampton Women Survey, UK (*SWS*) (*n* = 3,007) [58].

### Data access and federated analysis on DataSHIELD

In this study, we used pseudonymised data stored on local secure data servers in their original location [59,60,61,62] and harmonised according to protocols in the EU Child Cohort Network [39]. Cohort-specific description about methods for ascertaining and defining variables are documented in the EU Child Cohort Network catalogue (https://data-catalogue. molgeniscloud.org/catalogue/catalogue/#/) and the Maelstrom Catalogue (http://maelstrom-research.org) for studies in LifeCycle and EUCAN-Connect, respectively. Data were analysed remotely through the R-based and open-source software, DataSHIELD, which allows federated analysis through one-stage and two-stage IPD meta-analysis approaches with active disclosure controls [63,64,65,66]. Fourteen cohorts gave permission to analyse their data via Data-SHIELD, and two cohorts (AOF, CHILD) via data transfer agreements.

### Gestational age at birth

Information on GA (in days) was available as harmonised IPD with source of delivery information obtained from medical records in the majority of cohorts (S1 Table and S1 Text). GA was rounded to completed weeks and further categorised into five groups [67]: 28 to 33 weeks (very preterm), 34 to 36 weeks (late preterm), 37 to 38 weeks (early term), 39 to 41 weeks (full term), and 42 to 43 weeks (postterm).

## Offspring BMI and overweight and obesity

Information on height (cm) and weight (kg) was available as harmonised IPD measured in either a clinical setting or self-reported by parents or index child (S1 Table). BMI was calculated as weight (kg)/(height (m))$^2$ [68], and sex-and-age specific BMI z-scores were calculated per month using instructions from Vidmar and colleagues [69] and following the growth standard [70] and reference [71] from the World Health Organization (WHO). We defined overweight (including obesity) following WHO cutoffs, separately for children <5 years (>2 standard deviations above WHO Child Growth Standard median) and ≥5 years (>1 standard deviation above WHO Growth Reference median). In several cohorts (ALSPAC, BiB, DNBC, GEN R, INMA, NINFEA, NFBC1986, the Raine Study, SWS), multiple measurements of BMI were available for the same child within one or more of the six age-groups. In such cases, the latest available measurement within each age group was chosen.

## Confounders

Confounders were selected a priori as factors that were known or plausible causes of variation in GA and subsequent body size with a directed acyclic graph used in discussions to select the final set of confounders (S1 Fig).

The resulting confounders were as follows: maternal education (ISCED-2011/97, low/ medium/high) (S1 Text) [72]; maternal height (continuous, m); maternal prepregnancy BMI (continuous, kg/m$^2$); maternal smoking during pregnancy (yes/no); maternal age at child's birth (continuous, years); gestational diabetes (yes/no); gestational hypertension (yes/no); preeclampsia (yes/no); maternal ethnic background (western/nonwestern/mixed) (S1 Text); and parity (nulliparous/parous). For the objective of this study, we did neither include birth weight or puberty as they may distort interpretation of the results being intermediate variables on the causal pathway [25].

## Statistical analysis

Distributions of GA at birth, body size measures, and confounders were obtained for each cohort separately and for all cohorts combined.

We conducted a two-stage meta-analysis to estimate associations between GA with BMI and overweight, adjusted for confounders. We fitted a linear regression model to examine the associations of GA in weeks and in clinical categories with BMI z-scores. Models were fitted in each cohort separately, and cohort-specific coefficients and standard errors were combined and assigned weights using random-effects model to attain overall effect estimates [66,73]. The analyses were performed separately for the six age-groups (>0.0 to 0.5 years, >0.5 to 2.0 years, >2.0 to 5.0 years, >5.0 to 9.0 years, >9.0 to 14.0 years, and >14.0 to 19.0 years). To examine the associations between GA in weeks and in clinical categories and odds of overweight (compared with normal weight), we used a binomial logistic regression model.

The main results are those from regression analyses adjusted for the maximum set of baseline confounders available within each cohort. Models were adjusted for maternal age at child's birth, height, education, prepregnancy BMI, and parity in all cohorts. Models were additionally adjusted for maternal ethnic background, gestational hypertension, gestational diabetes, and preeclampsia in cohorts where these were available (Table 1).

Results are presented with 95% confidence intervals (CIs) and I$^2$ statistics [74]. We examined between-study heterogeneity by meta-regression in meta-analyses with considerable heterogeneity reflected by either I$^2$ > 75% or I$^2$ approximately 75% with effect estimates in different directions. The meta-regressions were conducted to determine which study characteristics were independently associated with between-study heterogeneity. In addition, we

**Table 1. Baseline characteristics of study participants in the 16 participating cohorts.**

| | Study population | Year of birth | Sex (%), female | GA at birth (weeks), mean (SD) | Maternal age at birth (years), mean (SD) | Maternal education (%), low | Maternal education (%), medium | Maternal education (%), high | Maternal ethnic background (%), western | Maternal ethnic background (%), nonwestern | Maternal ethnic background (%), mixed | Maternal height at birth (cm), mean (SD) | Prepregnancy BMI (kg/m$^2$), mean (SD) | Prepregnancy overweight (%) | Maternal smoking in pregnancy (%) | Gestational diabetes (%) | Gestational hypertension (%) | Maternal preeclampsia (%) | Parity, nulliparity (%) |
|---|---|---|---|---|---|---|---|---|---|---|---|---|---|---|---|---|---|---|---|
| **All cohorts (N = 16)** | **253,810** | **1985–2017** | **49.1** | **39.8 (1.8)** | **30.0 (4.7)** | **16.9** | **30.5** | **52.6** | **77.6** | **18.7** | **3.7** | **167.1 (6.5)** | **23.7 (4.3)** | **28.6** | **18.0** | **2.0** | **7.4** | **1.6** | **47.7** |
| **ALSPAC, United Kingdom** | 10,452 | 1991–1993 | 49.6 | 39.9 (1.7) | 28.8 (4.6) | 14.5 | 68.9 | 16.7 | 98.4 | 1.6 | 0.0 | 164.1 (6.7) | 22.6 (4.4) | 20.5 | 25.4 | 0.5 | 14.7 | 1.9 | 45.4 |
| **AOF, Canada** | 2,263 | 2008–2011 | 47.4 | 38.9 (1.7) | 31.3 (4.4) | 77.5 | 20.2 | 2.3 | 81.7 | 16.7 | 1.6 | 165.9 (7.0) | 24.5 (5.2) | 34.8 | 10.7 | 5.1 | 7.4 | 6.8 | 50.8 |
| **BiB, United Kingdom** | 13,097 | 2007–2011 | 48.4 | 39.5 (1.8) | 27.6 (5.6) | 56.8 | 15.8 | 27.4 | 42.2 | 55.9 | 1.9 | 161.6 (6.5) | 26.0 (5.6) | 50.2 | 16.3 | 8.0 | 7.1 | 2.6 | 39.5 |
| **CHILD, Canada** | 2,984 | 2009–2012 | 47.5 | 39.5 (1.4) | 31.8 (4.6) | 8.3 | 28.7 | 63.0 | 73.3 | 20.6 | 6.1 | 165.0 (6.9) | 24.2 (5.0) | 32.5 | 18.7 | 4.2 | 7.6 | 1.3 | 37.5 |
| **DNBC, Denmark** | 81,117 | 1996–2003 | 49.6 | 39.9 (1.8) | 30.1 (4.2) | 26.4 | 21.9 | 51.7 | | | | 168.8 (6.1) | 23.6 (4.2) | 27.4 | 25.4 | 0.9 | 12.6 | 2.4 | 47.6 |
| **EDEN, France** | 1,765 | 2003–2006 | 48.0 | 39.7 (1.7) | 29.6 (4.8) | 6.1 | 38.8 | 55.1 | 99.0 | 0.5 | 0.5 | 163.6 (6.2) | 23.2 (4.6) | 25.9 | 25.5 | 6.5 | 1.9 | 2.6 | 45.3 |
| **ELFE, France** | 15,506 | 2011 | 48.8 | 39.6 (1.5) | 30.4 (4.9) | 7.2 | 32.9 | 59.9 | 81.1 | 12.0 | 6.9 | 165.0 (6.3) | 23.4 (4.8) | 26.6 | 19.3 | 6.9 | 2.0 | 1.5 | 46.1 |
| **G21, Portugal** | 6,439 | 2005–2006 | 48.6 | 39.2 (1.6) | 29.3 (5.4) | 46.0 | 28.8 | 25.3 | 95.5 | 2.6 | 2.0 | 160.8 (6.2) | 24.0 (4.3) | 31.0 | 37.5 | 6.5 | 1.9 | 2.0 | 58.0 |
| **GECKO, the Netherlands** | 2,768 | 2006–2008 | 49.6 | 39.8 (1.6) | 30.7 (4.4) | 35.6 | 28.6 | 35.8 | 95.7 | 2.7 | 1.7 | 171.6 (6.3) | 24.7 (4.7) | 37.3 | 15.6 | 2.5 | 7.9 | 2.6 | 40.5 |
| **GEN R, the Netherlands** | 8,641 | 2002–2006 | 49.6 | 40.3 (1.8) | 30.6 (5.2) | 10.3 | 44.6 | 45.0 | 56.8 | 33.3 | 9.9 | 167.5 (7.5) | 23.6 (4.3) | 27.8 | 26.0 | 1.1 | 3.8 | 2.2 | 55.3 |
| **INMA, Spain** | 1,936 | 2004–2008 | 48.6 | 39.9 (1.5) | 31.8 (4.2) | 24.1 | 40.9 | 35.0 | 95.6 | 4.4 | 0.0 | 162.8 (6.2) | 23.5 (4.2) | 25.1 | 31.5 | 4.4 | | | 55.5 |
| **MoBa, Norway** | 86,553 | 1999–2009 | 48.7 | 39.8 (1.8) | 30.3 (4.5) | 2.2 | 31.8 | 66.0 | | | | 168.2 (5.9) | 24.0 (4.2) | 31.0 | 8.7 | 0.4 | 4.9 | 0.1 | 47.7 |
| **NFBC1986, Finland** | 8,325 | 1985–1986 | 49.1 | 39.8 (1.6) | 27.8 (5.5) | 38.9 | 37.3 | 23.7 | | | | 163.2 (5.5) | 22.3 (3.5) | 16.9 | 23.8 | | 4.0 | 2.5 | 34.2 |
| **NINFEA, Italy** | 6,515 | 2005–2017 | 49.3 | 39.7 (1.7) | 33.2 (4.3) | 4.6 | 32.5 | 62.9 | | | | 165.0 (6.2) | 22.5 (3.8) | 19.0 | 8.0 | 8.1 | 3.3 | 2.2 | 72.7 |
| **The Raine study, Australia** | 2,443 | 1989–1992 | 48.7 | 39.2 (2.0) | 27.8 (5.8) | 52.7 | 27.1 | 20.2 | 89.4 | 10.6 | 0.0 | 163.6 (6.6) | 22.3 (4.2) | 17.7 | 27.5 | 1.8 | 19.3 | 4.9 | 47.7 |
| **SWS, United Kingdom** | 3,007 | 1999–2007 | 48.0 | 39.7 (1.8) | 30.2 (3.8) | 12.0 | 59.3 | 28.7 | 95.7 | 3.7 | 0.6 | 163.3 (6.5) | 25.2 (4.8) | 41.3 | 15.6 | 1.3 | 3.3 | 2.8 | 51.5 |

Percentages include non-missing, and empty cells represent no available data.
BMI, body mass index; SD, standard deviation.

undertook "leave-one-out" analysis for cross-validation to explore the influence of each study on the overall estimate [75], while subgroup analysis with sex (boys versus girls), maternal education (high versus low/medium), maternal smoking in pregnancy (no versus yes), and birth year (<2001, ≥2001) was performed to measure the robustness of our findings.

Statistical analyses were performed using DataSHIELD and the Statistical Software R (v4.1) [76]. We mainly used the ds.getWGSR and ds.glmSLMA functions from the dsBaseClient (v6.1.0, https://github.com/datashield/dsBaseClient/) and the dh.makeStrata function from the ds-Helper package (https://github.com/lifecycle-project/ds-helper), as well as to the rma-package (v3.0.2) [77]. Forest plots were created using Excel 2016.

## Ethical approval

This study builds upon a federated analysis solution that facilitates cross-border sharing of harmonised and pseudonymised data in compliance with both European and national data protection, patient's rights, and research ethics regulation. Rather than sharing or transferring individual-level data, only nondisclosive low-dimensional summary statistics are made accessible upon request. Prior to analysing the data via DataSHIELD, the lead author obtained a waiver of consent (data access or data transfer agreement) from each cohort study via the partner in LifeCycle and EUCAN-Connect. The data from the 16 cohorts were only accessible as meta-data controlled through active disclosure protection, which mitigates the risk of identification of study participants. Further information about the infrastructural setup using DataSHIELD, and previous international networks using similar methods, are describe in details elsewhere [78,79,80].

Each participating cohort obtained a written consent from the mother/parents, while details about the study-specific ethical approvals are listed here in an alphabetic order after the cohort name: (ALSPAC) Ethical approval for the study was obtained from the ALSPAC Ethics and Law Committee and the Local Research Ethics Committees; (AOF) The All Our Families study was approved by the Child Health Research Office and the Conjoint Health Research Ethics Board of the Faculties of Medicine, Nursing, and Kinesiology, University of Calgary, and the Affiliated Teaching Institutions (Ethics ID 20821 and 22821); (BiB) Ethics approval has been obtained for the main platform study and all of the individual substudies from the Bradford Research Ethics Committee; (CHILD) Ethics approvals for the study were obtained at recruitment and at each data collection phase from all four Canadian sites; (DNBC) The DNBC complies with the Declaration of Helsinki and was approved by the Danish National Committee on Biomedical Research Ethics; (EDEN): The study received approval from the ethics committee (CCPPRB) of Kremlin Bicêtre on 12 December 2002 and from CNIL (Commission Nationale Informatique et Liberté), the French data privacy institution; (ELFE): Ethical approvals for data collection in maternity units and for each data collection wave during follow-up were obtained from the national advisory committee on information processing in health research (CCTIRS: Comité Consultatif sur le Traitement de l'Information en matière de Recherche dans le domaine de la Santé), the national data protection authority (CNIL: Comission Nationale Informatique et Liberté) and, in case of invasive data collection such as biological sampling, the committee for protection of persons engaged in research (CPP: Comité de Protection des Personnes). The ELFE study was also approved by the national committee for statistical information (CNIS: Conseil National de l'Information Statistique); (G21) Generation XXI was approved by the Portuguese Data Protection Authority and by the Ethics Committee of Hospital São João, and data confidentiality and protection were guaranteed in all procedures according to the Declaration of Helsinki. Signed informed consent was obtained for all adults and children participants had it signed by their legal guardian at every study

waves; (GECKO) The GECKO Drenthe study complies with the Declaration of Helsinki and was approved by the Medical Ethics Committee of the University Medical Center Groningen; (GEN R) The general design, all research aims and the specific measurements in the Generation R Study have been approved by the Medical Ethical Committee of the Erasmus Medical Center, Rotterdam. New measurements will only be embedded in the study after approval of the Medical Ethical Committee, (INMA) The INMA project was approved by the ethics committee in each area; (MoBa) The establishment and data collection in MoBa was previously based on a license from the Norwegian Data protection agency and approval from The Regional Committee for Medical Research Ethics, and it is now based on regulations related to the Norwegian Health Registry Act; (NINFEA) The Ethical Committee of the San Giovanni Battista Hospital and CTO/CRF/Maria Adelaide Hospital of Turin approved the NINFEA study (approval N. 0048362, and subsequent amendments); (The Raine study) The original cohort study received approval from the King Edward Memorial Hospital for women ethics committee in 1989 (DD/JS/459), and all subsequent follow-ups also received institutional human research ethics committee (HREC) approval prior to commencing. All participants were provided with participant information sheets and parents (Gen1) provided informed consent, and the child (Gen2) provided assent. When the Raine study Gen2 participants turned 18 years of age, ethics approval was further received from the University of Western Australia HREC (RA/4/1/2100) to contact and obtain consent from Raine study participants for any data that was collected before they were 18 to be used for future research. A further UWA HREC was provided to all a single 'overarching' approval code that recognises all previous approvals under which previous data and/or bio-samples were collected. This approval was received on 29 April 2020 and provides a single consolidated approval (RA/4/20/5722) for use of research data and/or bio-samples held in the Raine study data collection; (SWS) The study had full approval at each wave from the Southampton and Southwest Hampshire Local Research Ethics Committee.

## Results

### Descriptive statistics

A total of 253,810 mother–child dyads in 16 cohort studies from 11 countries had information on GA and at least one measurement of offspring BMI.

Descriptive information including characteristics of GA at birth, body size measures, and covariates for study participants are displayed separately for each cohort and for the cohorts combined in Tables 1–3.

There were distinct differences in the cohort-specific sample sizes (n = 1,765 to 86,553), distributions of maternal education (range: 2.2% to 77.5% for low), maternal ethnicity (range: 42.2% to 99.0% for western; 0.5% to 55.9% for nonwestern; 0.0% to 9.9% for mixed); maternal prepregnancy overweight (range: 16.9% to 50.2%), gestational hypertension (range: 1.9% to 19.3%), and parity (range for nulliparous: 34.2% to 72.7%) (Table 1).

The mean GA was 39.8 weeks, and overall 5.5% were born preterm (range: 3.1% to 7.5%), 17.8% (range: 11.8% to 31.6%) were born early term, 69.9% were born full term (range 61.1% to 73.6%), and 6.7% (range: 0.2% to 15.4%) were born postterm (Table 2). The majority of the cohorts had study participants included for analysis in all five categories of GA, except CHILD (34 to 43 weeks gestation).

From infancy to age 19 years, 711,856 measurements of BMI were available for 253,810 children. The number of cohorts and participants with data on BMI and overweight varied across the six age-bands with most cohorts and participants in infancy and mid-childhood and fewest in adolescence, where four cohorts (ALSPAC, DNBC, NFBC1986, and the Raine Study) contributed with data on 36,895 individuals. The proportion of children classified as overweight

**Table 2. Distribution of gestational age groups in the 16 participating cohorts.**

| | Gestational age at birth (completed weeks) | | | | |
|---|---|---|---|---|---|
| | Very preterm 28–33 weeks, n (%) | Late preterm, 34–36 weeks, n (%) | Early term, 37–38 weeks, n (%) | Full term, 39–41 weeks, n (%) | Post term, 42–43 weeks, n (%) |
| **All cohorts (N = 16)** | **3,137 (1.2)** | **11,061 (4.3)** | **45,088 (17.8)** | **177,465 (69.9)** | **17,059 (6.7)** |
| ALSPAC, United Kingdom | 112 (1.2) | 440 (4.2) | 1,779 (17.0) | 7,303 (69.9) | 808 (7.7) |
| AOF, Canada | 29 (1.3) | 114 (5.0) | 591 (26.1) | 1,525 (67.4) | 4 (0.2) |
| BiB, United Kingdom | 184 (1.4) | 607 (4.6) | 2,948 (22.5) | 9,189 (70.2) | 169 (1.3) |
| CHILD, Canada | | 126 (4.2) | 700 (23.5) | 2,133 (71.5) | 25 (0.8) |
| DNBC, Denmark | 1,028 (1.3) | 3,448 (4.3) | 13,286 (16.4) | 56,409 (69.5) | 6,946 (8.6) |
| EDEN, France | 25 (1.4) | 70 (4.0) | 335 (18.9) | 1,299 (73.6) | 36 (2.0) |
| ELFE, France | 70 (0.5) | 763 (4.9) | 3,195 (20.6) | 11,400 (73.5) | 78 (0.5) |
| G21, Portugal | 88 (1.4) | 372 (5.8) | 2,033 (31.6) | 3,935 (61.1) | 11 (0.2) |
| GECKO, the Netherlands | 18 (0.7) | 120 (4.3) | 551 (19.9) | 1,948 (70.4) | 131 (4.7) |
| GEN R, the Netherlands | 83 (1.0) | 302 (3.5) | 1,016 (11.8) | 5,911 (68.4) | 1,329 (15.4) |
| INMA, Spain | 8 (0.4) | 52 (2.7) | 373 (19.3) | 1,390 (71.8) | 113 (5.8) |
| MoBa, Norway | 1,215 (1.4) | 3,797 (4.4) | 14,415 (16.7) | 60,574 (70.0) | 6,552 (7.6) |
| NFBC1986, Finland | 104 (1.3) | 294 (3.5) | 1,455 (17.5) | 6,151 (73.9) | 321 (3.9) |
| NINFEA, Italy | 69 (1.1) | 293 (4.5) | 1,383 (21.2) | 4,510 (69.2) | 259 (4.0) |
| The Raine study, Australia | 50 (2.1) | 132 (5.4) | 509 (20.8) | 1,611 (65.9) | 141 (5.8) |
| SWS United Kingdom | 44 (1.5) | 131 (4.4) | 519 (17.3) | 2,177 (72.4) | 136 (4.5) |

Empty cells represent no available data.

also varied between cohorts and across age-bands due to different cutoffs used for children <5 years and in children ≥5 years (Table 3).

The percentage of missing values for baseline characteristics is presented in the Supporting information (S2 Table).

## Gestational age at birth and BMI z-scores

The overall unadjusted and adjusted estimates for the associations of GA in completed weeks and clinical categories with BMI z-score are displayed in Figs 1 and 2, while cohort-specific estimates are available in the Supporting information (S2 Fig).

**Table 3. Distribution and age of body size measurements in the 16 participating cohorts.**

| Cohort | Early Infancy >0–0.5 years (1–6 months) | | | | Late Infancy >0.5–2 years (7–24 months) | | | | Early Childhood >2–5 years (25–60 months) | | | | Mid-Childhood >5–9 years (61–108 months) | | | | Late Childhood >9–14 years (109–168 months) | | | | Adolescence >14–19 years (169–227 months) | | | |
|---|---|---|---|---|---|---|---|---|---|---|---|---|---|---|---|---|---|---|---|---|---|---|---|---|
| | N | Age in months (SD) | BMI z-score | Overweight, n (%) | N | Age in months (SD) | BMI z-score | Overweight, n (%) | N | Age in months (SD) | BMI z-score | Overweight, n (%) | N | Age in months (SD) | BMI z-score | Overweight, n (%) | N | Age in months (SD) | BMI z-score | Overweight, n (%) | N | Age in months (SD) | BMI z-score | Overweight, n (%) |
| All cohorts (N = 16) | 185,428 | 4.5 (1.3) | -0.19 | 3,804 (2.1) | 186,419 | 14.7 (4.0) | 0.36 | 9,979 (5.4) | 106,916 | 44.7 (10.5) | 0.34 | 6,512 (6.0) | 154,863 | 88.0 (10.8) | 0.17 | 21.1 | 78,230 | 136.7 (14.4) | 0.08 | 16,457 (21.0) | 54,155 | 211.8 (10.1) | 0.12 | 9,869 (18.2) |
| ALSPAC, United Kingdom | 1,014 | 3.8 (0.2) | -0.10 | 14 (1.4) | 1,368 | 17.2 (2.8) | 0.78 | 111 (8.1) | 1,261 | 47.1 (5.4) | 0.63 | 88 (7.0) | 8,298 | 97.9 (11.4) | 0.40 | 2,325 (28.0) | 9,202 | 156.5 (13.7) | 0.29 | 2,492 (27.1) | 7,703 | 206.5 (11.0) | 0.23 | 1,754 (22.8) |
| AOF, Canada | | | | | 1,412 | 17.2 (2.8) | 0.83 | 254 (18.0) | 1,878 | 44.1 (11.8) | 0.21 | 124 (6.6) | 1,745 | 92.9 (12.5) | 0.00 | 380 (21.8) | | | | | | | | |
| BiB, United Kingdom | 11,724 | 1.9 (1.1) | -0.59 | 68 (0.6) | 10,384 | 19.2 (5.5) | 0.19 | 536 (5.1) | 9,813 | 51.8 (9.2) | 0.48 | 839 (8.5) | 8,241 | 93.6 (12.7) | 0.29 | 2,377 (28.8) | 1,141 | 119.5 (6.6) | 0.37 | 397 (34.8) | | | | |
| CHILD, Canada | | | | | 2,847 | 12.7 (5.9) | 0.31 | 195 (6.8) | 2,750 | 46.5 (10.2) | 0.48 | 192 (7.0) | 1,652 | 61.9 (2.3) | 0.28 | 300 (18.2) | | | | | | | | |
| DNBC, Denmark | 51,411 | 5.2 (0.4) | -0.32 | 1,058 (2.1) | 51,429 | 12.4 (1.0) | 0.29 | 3,077 (6.0) | | | | | 42,930 | 84.1 (3.5) | 0.02 | 6,619 (15.4) | 44,070 | 136.3 (7.4) | -0.14 | 6,619 (15.0) | 38,351 | 216.3 (3.9) | 0.11 | 6,711 (17.5) |
| EDEN, France | 1,701 | 5.0 (0.8) | -0.28 | 20 (1.1) | 1,721 | 19.3 (4.8) | 0.35 | 76 (4.4) | 1,524 | 47.8 (8.0) | 0.12 | 43 (2.3) | 1,240 | 86.8 (13.8) | 0.05 | 202 (16.3) | 807 | 133.9 (9.2) | -0.10 | 160 (19.8) | | | | |
| ELFE, France | 13,876 | 3.8 (0.9) | -0.25 | 211 (1.5) | 13,628 | 14.2 (5.0) | 0.19 | 502 (3.7) | 12,002 | 46.4 (9.5) | 0.05 | 385 (3.2) | 7,812 | 86.7 (16.6) | -0.05 | 1,251 (16.0) | 492 | 109.3 (0.3) | -0.08 | 70 (14.2) | | | | |
| G21, Portugal | | | | | | | | | 5,048 | 51.2 (3.4) | 0.63 | 535 (10.6) | 5,604 | 85.2 (4.2) | 0.74 | 2,105 (37.6) | 4,884 | 121.9 (4.2) | 0.72 | 2,084 (42.7) | | | | |
| GECKO, the Netherlands | 2,714 | 5.0 (0.7) | -0.18 | 31 (1.1) | 2,683 | 17.3 (3.6) | 0.50 | 122 (4.5) | 2,247 | 41.9 (7.6) | 0.33 | 92 (4.1) | 2,258 | 70.4 (4.2) | 0.43 | 514 (22.8) | 2,178 | 127.5 (5.4) | 0.26 | 535 (24.6) | | | | |
| GEN R, the Netherlands | 6,252 | 4.4 (1.2) | -0.17 | 109 (1.7) | 6,973 | 17.8 (3.6) | 0.68 | 549 (7.9) | 6,452 | 42.1 (6.6) | 0.38 | 350 (5.4) | 6,609 | 74.3 (6.7) | 0.46 | 1,688 (25.5) | 5,591 | 117.5 (3.9) | 0.35 | 1,472 (26.3) | | | | |
| INMA, Spain | 1,873 | 4.9 (0.9) | -0.14 | 30 (1.6) | 1,927 | 17.6 (3.1) | 0.47 | 116 (6.0) | 1,611 | 52.7 (2.8) | 0.61 | 146 (9.1) | 1,417 | 93.5 (8.2) | 0.81 | 573 (40.4) | 937 | 130.8 (7.8) | 0.73 | 406 (43.3) | | | | |
| MOBA, Norway | 83,416 | 4.5 (1.3) | -0.06 | 1,961 (2.4) | 75,091 | 18.5 (4.2) | 0.37 | 3,210 (4.3) | 48,835 | 41.4 (11.1) | 0.35 | 2,982 (6.1) | 52,132 | 91.7 (9.7) | 0.11 | 10,808 (20.7) | | | | | | | | |
| NFBC1986, Finland | 5,671 | 5.2 (0.7) | 0.05 | 149 (2.6) | 5,795 | 18.5 (4.2) | 0.64 | 402 (6.9) | 5,470 | 49.8 (7.6) | 0.35 | 207 (3.8) | 8,081 | 92.6 (8.8) | 0.32 | 1,973 (24.4) | 5,318 | 157.3 (9.8) | 0.22 | 1,212 (22.8) | 6,535 | 195.0 (8.3) | -0.01 | 992 (15.2) |
| NINFEA, Italy | 5,036 | 4.7 (1.5) | -0.39 | 129 (2.6) | 6,032 | 17.2 (4.2) | 0.41 | 468 (7.8) | 4,721 | 51.2 (5.1) | 0.06 | 241 (5.1) | 2,724 | 86.8 (4.2) | 0.08 | 592 (21.7) | 1,005 | 131.2 (16.6) | 0.04 | 216 (21.5) | | | | |
| The Raine study, Australia | | | | | 2,220 | 13.9 (1.6) | 0.45 | 109 (4.9) | 584 | 26.2 (1.6) | 0.06 | 15 (2.6) | 2,107 | 93.9 (9.0) | 0.39 | 536 (25.4) | 1,732 | 133.8 (15.1) | 0.52 | 569 (32.9) | 1,566 | 196.3 (14.5) | 0.41 | 412 (26.3) |
| SWS, United Kingdom | 740 | 5.8 (0.1) | 0.21 | 24 (3.2) | 2,909 | 14.8 (5.2) | 0.69 | 252 (8.7) | 2,720 | 41.5 (7.5) | 0.53 | 173 (6.3) | 2,013 | 84.3 (10.2) | 0.29 | 451 (22.4) | 873 | 111.4 (2.6) | 0.19 | 225 (25.7) | | | | |

Percentages include non-missing, and empty cells represent no available data.

BMI, body mass index; N, sample size; SD, standard deviation.

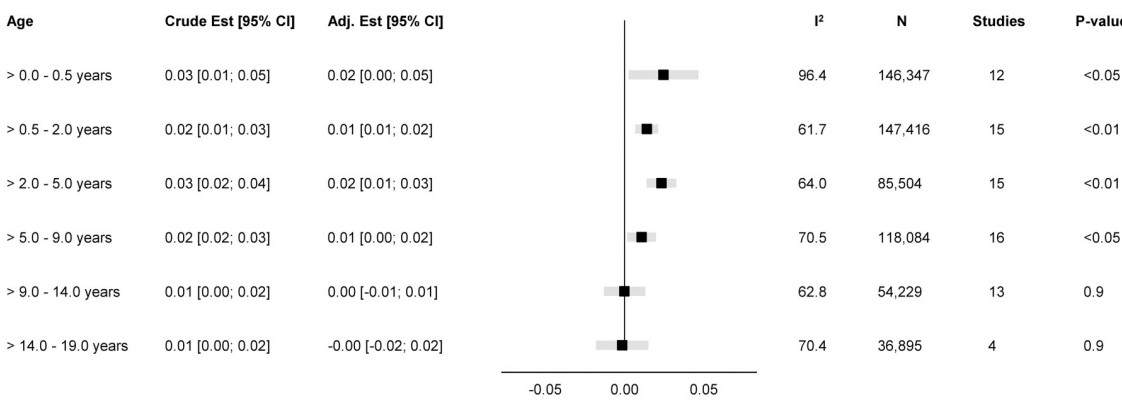

| Age | Crude Est [95% CI] | Adj. Est [95% CI] | | I² | N | Studies | P-value |
|---|---|---|---|---|---|---|---|
| > 0.0 - 0.5 years | 0.03 [0.01; 0.05] | 0.02 [0.00; 0.05] | | 96.4 | 146,347 | 12 | <0.05 |
| > 0.5 - 2.0 years | 0.02 [0.01; 0.03] | 0.01 [0.01; 0.02] | | 61.7 | 147,416 | 15 | <0.01 |
| > 2.0 - 5.0 years | 0.03 [0.02; 0.04] | 0.02 [0.01; 0.03] | | 64.0 | 85,504 | 15 | <0.01 |
| > 5.0 - 9.0 years | 0.02 [0.02; 0.03] | 0.01 [0.00; 0.02] | | 70.5 | 118,084 | 16 | <0.05 |
| > 9.0 - 14.0 years | 0.01 [0.00; 0.02] | 0.00 [-0.01; 0.01] | | 62.8 | 54,229 | 13 | 0.9 |
| > 14.0 - 19.0 years | 0.01 [0.00; 0.02] | -0.00 [-0.02; 0.02] | | 70.4 | 36,895 | 4 | 0.9 |

Mean body mass index z-score per week increase in gestational age

**Fig 1. Forest plot of associations between GA (completed weeks) and BMI z-score.** Overall unadjusted and adjusted estimates with 95% CIs from IPD meta-analyses of the study-specific linear regression models, where cohorts were assigned weights under the random-effects model. The dot in the forest plot represents the adjusted estimates, while the whiskers span the 95% CI, whereas $I^2$-statistics ($I^2$), sample size (N), studies, and $p$-value (two-sided, <0.05) relate to adjusted estimates. Estimates reflect a mean difference in BMI z-scores per week increase in gestational at birth in early infancy (>0–0.5 years), late infancy (>0.5–2.0 years), early childhood (>2–5 years), mid-childhood (>5–9 years), late childhood (>9–14 years), and adolescence (>14–19 years). Models are adjusted for sex of child, and the following maternal characteristics: age at child's birth, education, height, prepregnancy BMI, smoking during pregnancy, parity, ethnic background, gestational diabetes and hypertension, and preeclampsia. Cohort-specific estimates were adjusted for the maximum available set of the confounding variables (see Table 1). BMI, body mass index; CI, confidence interval; GA, gestational age; IPD, individual participant data.

The adjusted estimates indicate a positive association of GA with BMI in early infancy (>0.0 to 0.5 years): 0.02 SD per week increase in GA [95% CI: 0.00, 0.05, $p < 0.05$], GA in clinical categories was associated with a BMI z-score of −0.55 [95% CI: −0.82, −0.28, $p < 0.01$] for very preterm and −0.15 [95% CI: −0.26, −0.05, $p < 0.01$] for late preterm compared to full term. Results attenuated through childhood and continued to decrease to zero by adolescence (0.00 [95% CI: −0.02, 0.02], $p$ 0.9) with no difference in BMI z-score between preterm and full term peers.

Between-study heterogeneity was examined through meta-regression in four age-bands (>0.0 to 0.5 years, >5.0 to 9.0 years, >9.0 to 14.0 years, and >14.0 to 19.0 years) having considerable heterogeneity, with largest $I^2$-statistics (96.4%) in early infancy (S3 Table). We examined age at measurement, child sex, maternal education, and maternal smoking in pregnancy as between-study characteristics. The meta-regression found age at measurement to be significantly associated with heterogeneity in early infancy (β = −0.029, se = 0.008, $p < 0.01$); maternal education in late childhood (β 0.001, se 0.001, $p$ 0.05), and both maternal education (β = 0.001, se = 0.001, $p < 0.01$) and smoking in pregnancy (β 0.007, se 0.003, $p$ 0.01) in adolescence.

The "leave-one-out" analyses gave similar overall effect estimates in all age-bands and did not change between-study heterogeneity markedly (S3 Fig); however, in adolescence, leaving out ALSPAC changed the $I^2$ from 70.4% to 0.4% (S3F Fig).

Subgroup analyses were consistent with the main findings across sex (S4 Fig), maternal educational level (S5 Fig), and pregnancy smoking status (S6 Fig).

## Gestational age at birth and overweight

The overall unadjusted and adjusted estimates for the associations of GA in completed weeks and clinical categories with odds of overweight are displayed in Figs 3 and 4, while cohort-specific estimates are available in the Supporting information (S8 Fig).

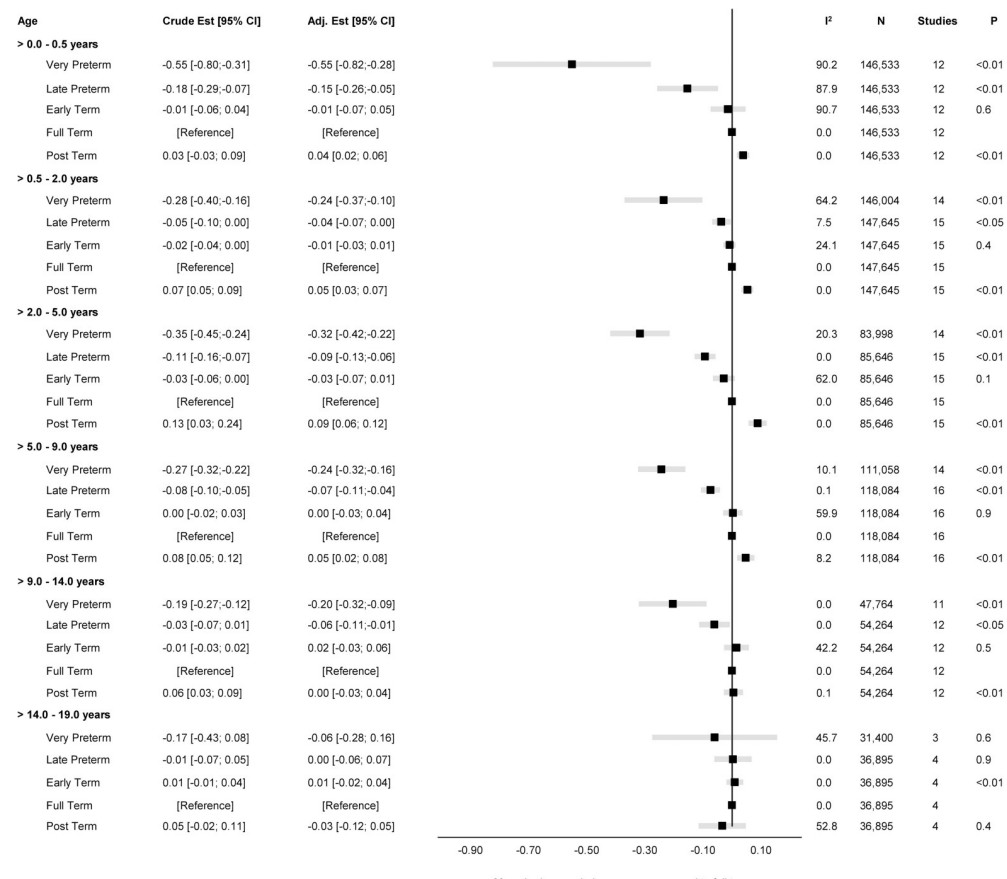

**Fig 2. Forest plot of associations between GA (clinical categories) and BMI z-score.** Overall unadjusted and adjusted estimates with 95% CIs from IPD meta-analyses of the study-specific linear regression models, where cohorts were assigned weights under the random-effects model. The dot in the forest plot represents the adjusted estimates, while the whiskers span the 95% CI, whereas $I^2$-statistics ($I^2$), sample size (N), studies, and $p$-value (two-sided, <0.05) relate to adjusted estimates. Estimates reflect a mean BMI z-scores compared to full term (reference category) in early infancy (>0–0.5 years), late infancy (>0.5–2.0 years), early childhood (>2–5 years), mid-childhood (>5–9 years), late childhood (>9–14 years), and adolescence (>14–19 years). Models are adjusted for sex of child, and the following maternal characteristics: age at child's birth, education, height, prepregnancy BMI, smoking during pregnancy, parity, ethnic background, gestational diabetes and hypertension, and preeclampsia. Cohort-specific estimates were adjusted for the maximum available set of the confounding variables (see Table 1). BMI, body mass index; CI, confidence interval; GA, gestational age; IPD, individual participant data.

There was a positive association of GA with odds of overweight (adj. OR 1.02 per week increase in GA) in late infancy [95% CI: 1.00, 1.03, *p 0.1*] and early childhood [95% CI: 0.99, 1.05, *p 0.2*]. Results attenuated through childhood and continued to decrease to below one by late childhood. In adolescence (>14.0 to 19.0 years), there was a negative association of GA with odds of overweight with very preterm having a significantly increased risk of overweight (adj. OR 1.46 [95% CI: 1.03, 2.08, *p* < 0.05] compared with full term peers.

None of the five age-bands had considerable between-study heterogeneity ($I^2$ < 55%); hence, we did not perform meta-regression for the associations of GA with odds of overweight. The "leave-one-out" analyses were consistent with the main findings without changing the overall effect estimate in any of the age-bands or any notable changes in the between-study heterogeneity (S9 Fig). The subgroup analysis showed no difference in the associations of GA with odds of overweight for sex (S10 Fig), maternal educational level (S11 Fig), pregnancy smoking status (S12 Fig), or year of birth (S13 Fig).

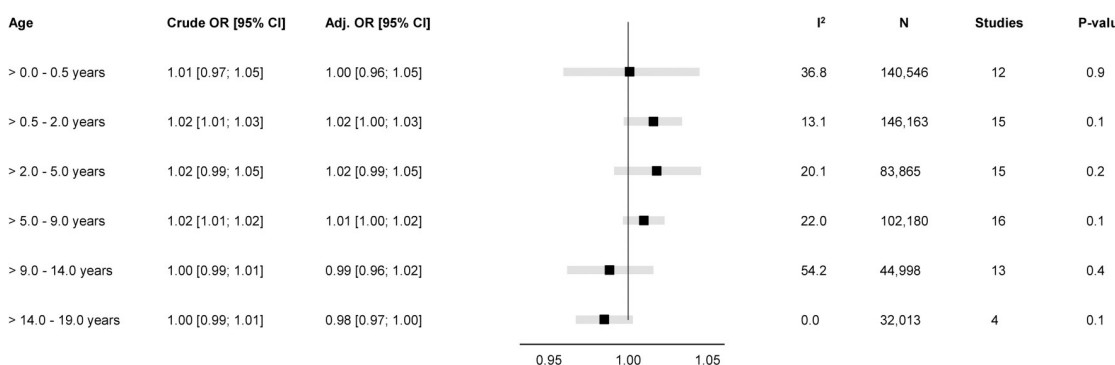

**Fig 3. Forest plot of associations between GA (completed weeks) and odds of overweight.** Overall unadjusted and adjusted ORs with 95% CIs from IPD meta-analyses of the study-specific logistic regression model, where cohorts were assigned weights under the random-effects model. The dot in the forest plot represents the adjusted estimates, while the whiskers span the 95% CI, whereas $I^2$-statistics ($I^2$), sample size (N), studies, and *p*-value (two-sided, <0.05) relate to adjusted estimates. Estimates reflect OR for overweight per week increase in gestational at birth in early infancy (>0–0.5 years), late infancy (>0.5–2.0 years), early childhood (>2–5 years), mid-childhood (>5–9 years), late childhood (>9–14 years), and adolescence (>14–19 years). Models are adjusted for sex of child, and the following maternal characteristics: age at child's birth, education, height, prepregnancy BMI, smoking during pregnancy, parity, ethnic background, gestational diabetes and hypertension, and preeclampsia. Cohort-specific estimates were adjusted for the maximum available set of the confounding variables (see Table 1). BMI, body mass index; CI, confidence interval; GA, gestational age; IPD, individual participant data; OR, odds ratio.

## Discussion

In this two-stage meta-analysis using IPD on 253,810 live-born singletons from 16 birth cohorts, we found a potentially important association in early infancy between GA and BMI, and in adolescence for the association of GA with odds of overweight. Difference in BMI z-score between categories of GA attenuated markedly after infancy throughout adolescence. A similar trend was observed for the association of GA with odds of overweight; however, by adolescence, increased odds of overweight was observed in very preterm compared with full term peers. Despite heterogeneity in cohort characteristics, our main findings were consistent across cohorts, and the supplementary analyses showed associations to be robust.

Previous studies [26,27,28,81,82] and a meta-analysis [35] have shown consistent results for the association between GA and BMI in childhood with lower BMI in preterm children compared with full term peers, although several methodological issues should be taken into account when interpreting these findings. In contrast, surprisingly few studies have examined the association of GA with later overweight, particularly in childhood.

Existing evidence for the association between GA and BMI rely on small sample sizes from different countries; different GA categorisation and reference group; and variations in use of BMI indices (z-scores or natural units, external or internal reference, IOTF or WHO reference). In our study, we observed a positive overall estimate for the association between GA and BMI in early infancy through mid-childhood with lower BMI z-score in very and late preterm compared to full term peers. In addition, we found age to be the main driver of between-study heterogeneity in early infancy suggesting that GA has a potentially important association in infancy. These findings are in line with descriptive results in infancy and early childhood from Australia [26] and Sweden [27]. In the Australian cohort, lower BMI z-scores were found among 225 extremely preterm compared with 253 term controls at both 2 and 5 years, while researchers in Sweden reported lower mean BMI z-scores at 2 and 5 years among 152 Swedish children born between 32 and 37 weeks compared with a large reference population. Our study showed a weaker association that most likely is explained by adjustment for confounders, but also methodological differences.

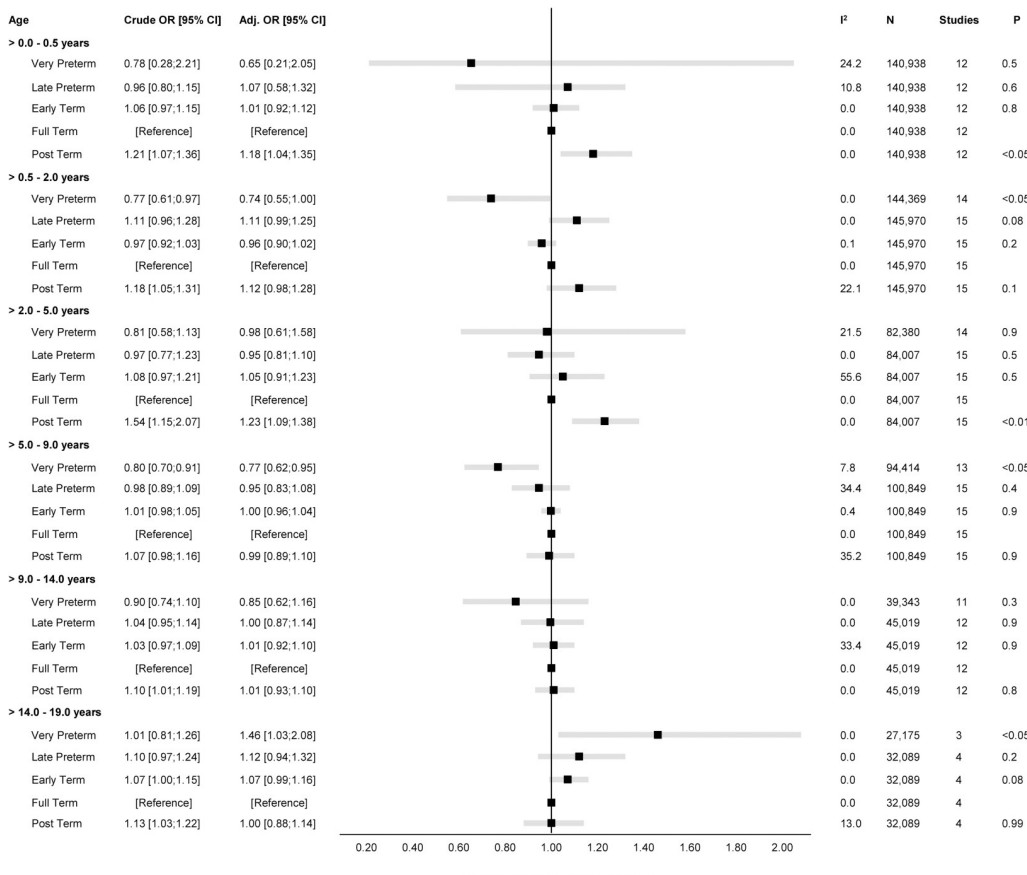

**Fig 4. Forest plot of associations between GA (clinical categories) and odds of overweight.** Overall unadjusted and adjusted ORs with 95% CIs from IPD meta-analyses of the study-specific logistic regression model estimates, where cohorts were assigned weights under the random-effects model. The dot in the forest plot represents the adjusted OR, while the whiskers span the 95% CI, whereas $I^2$-statistics ($I^2$), sample size (N), studies, and *p*-value (two-sided, <0.05) relate to adjusted OR. Estimates reflect OR for overweight compared to full term (reference category) in early infancy (>0–0.5 years), late infancy (>0.5–2.0 years), early childhood (>2–5 years), mid-childhood (>5–9 years), late childhood (>9–14 years), and adolescence (>14–19 years). Models are adjusted for sex of child, and the following maternal characteristics: age at child's birth, education, height, prepregnancy BMI, smoking during pregnancy, parity, ethnic background, gestational diabetes and hypertension, and preeclampsia. Cohort-specific estimates were adjusted for the maximum available set of the confounding variables (see Table 1). BMI, body mass index; CI, confidence interval; GA, gestational age; IPD, individual participant data; OR, odds ratio.

Our analyses revealed that the overall associations between GA and BMI attenuated in mid- and late childhood, but very and late preterm children remained at a lower BMI compared to their full term counterparts. This is supported by findings from studies in both Brazil and the United Kingdom across age 6 to 12 years [28,31,81]. By adolescence, we found no difference in BMI across categories of GA, and this is in line with previous findings from the study in Australia, and the two recent studies where preterm individuals in Brazil (≤33 weeks, 34 to 36 weeks) and the UK (≤25 weeks) reached similar BMI as their term counterparts at the age 18 to 19 years [26,28,31]. Hence, our study findings and previous evidence indicate that the overall association between GA and BMI attenuate through childhood with even the most preterm reaching similar BMI as their term counterparts by adolescence [20,26].

A rapid phase of growth has been proposed to evolve into increased susceptibility of later overweight [21,83,84,85], but only few studies have examined the relationship between GA and later overweight in childhood or adulthood [10,29,30,86]. The overall effect estimates

from our main analysis showed a weak association between GA and overweight from early infancy through mid-childhood with only very preterm in mid-childhood being at lower odds of overweight than full term peers. In contrast, a cohort study from Chile based on 153,635 children aged 6 to 8 years reported that term children are a lower risk of overweight (OR 0.84 [95% CI: 0.79, 0.88]) than preterm peers (reference group, (≤37 weeks) [29]). However, as highlighted by the authors, a major limitation of their study was the lack of information on obstetric maternal characteristics and maternal prepregnancy BMI.

In accordance with a cohort study from the UK on 11,765 children aged 11 years [30], we found no difference in odds of overweight between preterm and full term children in late childhood (>9.0 to 14.0 years).

Our study extends previous research by examining the association between GA and overweight in adolescence and across key stages of growth development throughout childhood. Moreover, our study design and large sample size enables an examination of odds of overweight in preterm adolescents and provides insights about this association across a wide range of GA. This distinction between degrees of preterm births is important as shorter gestational duration is associated with increased risk of mortality, disability, and morbidity across the life span [87]. Also, considering preterm births as not being homogeneous in causes and consequences was highlighted by others [88,89] as an important approach when interpreting such results, but a major limitation in current evidence [6,7,34].

Our main analysis suggested that very preterm may have an increased odds of overweight in adolescence compared with full term peers. Despite heterogeneity in characteristics for the four cohorts (ALSPAC, DNBC, NFBC1986, and the Raine Study) included for this age-band, our supplementary analyses addressed robustness in the findings. Our results are further supported by findings from two comparable studies conducted in Finland and Australia, where an increased odds of overweight was reported in preterm individuals aged 23 and 35 years, respectively [86,90].

In summary, this study sheds new light on factors influencing BMI and risk of developing overweight from infancy through adolescence. Furthermore, adding to the evidence within the domain of developmental origins of health and disease (DOHaD) with studies mainly using birth weight as proxy of prematurity, we used information on actual length of gestation and across a wide range of GA [23,91]. Our analysis revealed that, although preterm infants are relatively small at birth, they reach similar BMI and odds of overweight as term peers in adolescence. The underlying mechanisms from the current observational data are unknown. However, in accordance with previous findings, our pattern of results suggests that preterm infants may be at an increased odds of overweight later in life, even though BMI in preterm and full term is similar. In addition, it should be noted that mediating exposures such as birth weight, congenital anomalies, and breast feeding practices may also affect the relationship between GA and later body size.

An important strength in the current study is the large sample size with information on more than 250,000 mother–child dyads from 16 prospective pregnancy and birth cohorts in Europe, North America, and Australasia. We used comprehensive obstetric and maternal data as well as multiple BMI measurements following birth through adolescence, which allowed us to adjust analyses. Additionally, the large sample size enabled us to assess associations successively using clinical categories of preterm birth to age 19 years. We also examined the robustness of our findings performing several sensitivity analyses. Furthermore, the federated analysis approach using DataSHIELD proves a key advantage since it enables identical and reproducible analysis across multiple cohorts [39,92,93].

The limitations include considerable variations in both measurement and availability of exposures, covariates, and outcomes. However, this was explored by meta-regressions on

multiple covariates showing that study characteristics were independently associated with between-study heterogeneity only in the associations of GA with BMI. Age at measurement was the main contributor to heterogeneity in early infancy, but not in childhood and adolescence. This suggests that GA is important for BMI in early life but attenuates consistently as children get older. In late childhood and adolescence, maternal education and maternal smoking in pregnancy were independently associated with the observed heterogeneity. The method used to measure growth differed between cohorts, but was not explored any further, although it might be relevant [94].

Residual confounding may be another limitation in this study as the confounders are harmonised across studies, which gives the lowest common denominator. Also, several large cohorts (DNBC, MoBa, NFBC1986, the Raine study) had no available information on maternal ethnic background, which could bias our results. However, we had reports that the cohorts were homogeneous (>95% western) [45,52,53,56]; hence, we do not assume this affected our findings. Moreover, we did not deal with missingness as imputation and other methods were under development in DataSHIELD at time of the study, but we acknowledge that missing data on covariates may bias our findings, yet it is difficult to say in what direction.

As survival rates and postnatal treatment for preterm infants have improved in the last 20 years, while the global burden of obesity has increased [3,22,95], distribution of GA and proportion of overweight in the earliest cohorts are likely to differ from that in populations born more recently, with the former potentially being more selected and healthy later [34,96]. We found, however, no difference in the stratified analyses by year of birth (S7 and S13 Figs).

Our study is based on data from high-income countries; hence, the findings may not be generalisable to settings in low- and middle-income countries with higher estimates of preterm birth and rapid nutritional transition [4,95]. The proportion of preterm births was low in the participating cohorts and, despite being consistent with the global estimates for preterm births in Europe, one cohort recruited individuals in the third trimester, which may have led to exclusion of some individuals.

Lastly, for the objective of this study, we did not explore the role of mediating factors, such as size for GA, feeding practices, or puberty, although these may play a role in the associations observed [97,98,99].

In conclusion, based on data from infancy through adolescence in 16 cohort studies, we found that GA is important for growth in infancy, but the strength of association attenuated consistently with age. By adolescence, preterm individuals have on average a similar mean BMI to peers born at term.

## Supporting information

**S1 Fig. Directed acyclic graph for the association between gestational age at birth and body size.**
(TIF)

**S2 Fig. Forest plot of cohort-specific associations between GA (in weeks) and BMI z-score.**
Unadjusted and adjusted estimates with 95% CIs from study-specific linear regression models, where cohorts were assigned weights under the random-effects model to attain the overall estimates. The dot in the forest plot represents the adjusted estimates, while the whiskers span the 95% CI, whereas $I^2$-statistics ($I^2$), sample size (N), studies, and *p*-value (two-sided, <0.05) relate to adjusted estimates. Estimates reflect mean differences in BMI z-score per week increase in GA at birth in (**A**) early infancy (>0.0–0.5 years), (**B**) late infancy (>0.5–2.0 years), (**C**) early childhood (>2.0–5.0 years), (**D**) mid-childhood (>5.0–9.0 years), (**E**) late childhood (>9.0–14.0 years), and (**F**) adolescence (>14.0–19.0 years). Models are adjusted for sex of

child, and the following maternal characteristics: age at child's birth, education, height, pre-pregnancy BMI, smoking during pregnancy, parity, ethnic background, gestational diabetes and hypertension, and preeclampsia. Cohort-specific estimates were adjusted for the maximum available set of the confounding variables (see Table 1). BMI, body mass index; CI, confidence interval; GA, gestational age.

(TIF)

**S3 Fig. Forest plot of 'leave-one-out' analysis for the association between GA (in weeks) and BMI z-score.** Overall adjusted estimates of BMI z-score with 95% CIs from study-specific linear regression models, where cohorts were assigned weights under the random-effects model to attain the estimate. The dot in the forest plot represents the adjusted estimates, while the whiskers span the 95% CI, whereas $I^2$-statistics ($I^2$), sample size (N), studies, and $p$-value (two-sided, $<0.05$) relate to adjusted estimates. Estimates reflect mean differences in BMI z-score per week increase in GA at birth in (**A**) early infancy ($>0.0–0.5$ years), (**B**) late infancy ($>0.5–2.0$ years), (**C**) early childhood ($>2.0–5.0$ years), (**D**) mid-childhood ($>5.0–9.0$ years), (**E**) late childhood ($>9.0–14.0$ years), and (**F**) adolescence ($>14.0–19.0$ years). Models are adjusted for sex of child, and the following maternal characteristics: age at child's birth, education, height, prepregnancy BMI, smoking during pregnancy, parity, ethnic background, gestational diabetes and hypertension, and preeclampsia. Cohort-specific estimates were adjusted for the maximum available set of the confounding variables (see Table 1). BMI, body mass index; CI, confidence interval; GA, gestational age.

(TIF)

**S4 Fig. Forest plot of associations between GA (in weeks) and BMI z-score by sex.** Overall adjusted estimates of BMI z-score with 95% CIs from study-specific linear regression models, where cohorts were assigned weights under the random-effects model to attain the estimate. The blue (male) and yellow (female) dots in the forest plot represent the adjusted estimates, while the whiskers span the 95% CI, whereas $I^2$-statistics ($I^2$), sample size (N), studies, and $p$-value (two-sided, $<0.05$) relate to adjusted estimates. Estimates reflect mean differences in BMI z-score per week increase in GA at birth in early infancy ($>0.0–0.5$ years), late infancy ($>0.5–2.0$ years), early childhood ($>2.0–5.0$ years), mid-childhood ($>5.0–9.0$ years), late childhood ($>9.0–14.0$ years), and adolescence ($>14.0–19.0$ years). Models are adjusted for sex of child, and the following maternal characteristics: age at child's birth, education, height, pre-pregnancy BMI, smoking during pregnancy, parity, ethnic background, gestational diabetes and hypertension, and preeclampsia. Cohort-specific estimates were adjusted for the maximum available set of the confounding variables (see Table 1). BMI, body mass index; CI, confidence interval; GA, gestational age.

(TIF)

**S5 Fig. Forest plot of associations between GA (in weeks) and BMI z-score by maternal education.** Overall adjusted estimates of BMI z-score with 95% CIs from study-specific linear regression models, where cohorts were assigned weights under the random-effects model to attain the estimate. The blue (high educational level) and yellow (low/medium educational level) dots in the forest plot represent the adjusted estimates, while the whiskers span the 95% CI, whereas $I^2$-statistics ($I^2$), sample size (N), studies, and $p$-value (two-sided, $<0.05$) relate to adjusted estimates. Estimates reflect mean differences in BMI z-score per week increase in GA at birth in early infancy ($>0.0–0.5$ years), late infancy ($>0.5–2.0$ years), early childhood ($>2.0–5.0$ years), mid-childhood ($>5.0–9.0$ years), late childhood ($>9.0–14.0$ years), and adolescence ($>14.0–19.0$ years). Models are adjusted for sex of child, and the following maternal characteristics: age at child's birth, education, height, prepregnancy BMI, smoking during

pregnancy, parity, ethnic background, gestational diabetes and hypertension, and preeclampsia. Cohort-specific estimates were adjusted for the maximum available set of the confounding variables (see Table 1). BMI, body mass index; CI, confidence interval; GA, gestational age. (TIF)

**S6 Fig. Forest plot of associations between GA (in weeks) and BMI z-score by maternal smoking in pregnancy.** Overall adjusted estimates of BMI z-score with 95% CIs from study-specific linear regression models, where cohorts were assigned weights under the random-effects model to attain the estimate. The blue (smoking in pregnancy) and yellow (no smoking in pregnancy) dots in the forest plot represent the adjusted estimates, while the whiskers span the 95% CI, whereas $I^2$-statistics ($I^2$), sample size (N), studies, and $p$-value (two-sided, $<0.05$) relate to adjusted estimates. Estimates reflect mean differences in BMI z-score per week increase in GA at birth in early infancy ($>0.0–0.5$ years), late infancy ($>0.5–2.0$ years), early childhood ($>2.0–5.0$ years), mid-childhood ($>5.0–9.0$ years), late childhood ($>9.0–14.0$ years), and adolescence ($>14.0–19.0$ years). Models are adjusted for sex of child, and the following maternal characteristics: age at child's birth, education, height, prepregnancy BMI, smoking during pregnancy, parity, ethnic background, gestational diabetes and hypertension, and preeclampsia. Cohort-specific estimates were adjusted for the maximum available set of the confounding variables (see Table 1). BMI, body mass index; CI, confidence interval; GA, gestational age. (TIF)

**S7 Fig. Forest plot of associations between GA (in weeks) and BMI z-score by year of birth.** Overall adjusted estimates of BMI z-score with 95% CIs from study-specific linear regression models, where cohorts were assigned weights under the random-effects model to attain the estimate. The blue ($<2001$) and yellow ($\geq2001$) dots in the forest plot represent the adjusted estimates, while the whiskers span the 95% CI, whereas $I^2$-statistics ($I^2$), sample size (N), studies, and $p$-value (two-sided, $<0.05$) relate to adjusted estimates. Estimates reflect mean differences in BMI z-score per week increase in GA at birth in early infancy ($>0.0–0.5$ years), late infancy ($>0.5–2.0$ years), early childhood ($>2.0–5.0$ years), mid-childhood ($>5.0–9.0$ years), late childhood ($>9.0–14.0$ years), and adolescence ($>14.0–19.0$ years). Models are adjusted for sex of child, and the following maternal characteristics: age at child's birth, education, height, prepregnancy BMI, smoking during pregnancy, parity, ethnic background, gestational diabetes and hypertension, and preeclampsia. Cohort-specific estimates were adjusted for the maximum available set of the confounding variables (see Table 1). BMI, body mass index; CI, confidence interval; GA, gestational age. (TIF)

**S8 Fig. Forest plot of cohort-specific associations between GA (in weeks) and odds of overweight.** Unadjusted and adjusted ORs with 95% CIs from study-specific logistic regression models, where cohorts were assigned weights under the random-effects model to attain the overall estimates. The dot in the forest plot represents the adjusted OR, while the whiskers span the 95% CI, whereas $I^2$-statistics ($I^2$), sample size (N), studies, and $p$-value (two-sided, $<0.05$) relate to adjusted OR. Estimates reflect OR of overweight (vs. normal weight) per week increase in GA at birth in (**A**) early infancy ($>0.0–0.5$ years), (**B**) late infancy ($>0.5–2.0$ years), (**C**) early childhood ($>2.0–5.0$ years), (**D**) mid-childhood ($>5.0–9.0$ years), (**E**) late childhood ($>9.0–14.0$ years), and (**F**) adolescence ($>14.0–19.0$ years). Models are adjusted for sex of child, and the following maternal characteristics: age at child's birth, education, height, prepregnancy BMI, smoking during pregnancy, parity, ethnic background, gestational diabetes and hypertension, and preeclampsia. Cohort-specific estimates were adjusted for the

maximum available set of the confounding variables (see Table 1). BMI, body mass index; CI, confidence interval; GA, gestational age; OR, odds ratio.
(TIF)

**S9 Fig. Forest plot of "leave-one-out" analysis for the association between GA (in weeks) and odds of overweight.** Adjusted ORs with 95% CIs from study-specific logistic regression models, where cohorts were assigned weights under the random-effects model to attain the overall estimates. The dot in the forest plot represents the adjusted OR, while the whiskers span the 95% CI, whereas $I^2$-statistics ($I^2$), sample size (N), studies, and $p$-value (two-sided, $<$0.05) relate to adjusted OR. Estimates reflect OR of overweight (vs. normal weight) per week increase in GA at birth in (**A**) early infancy ($>$0.0–0.5 years), (**B**) late infancy ($>$0.5–2.0 years), (**C**) early childhood ($>$2.0–5.0 years), (**D**) mid-childhood ($>$5.0–9.0 years), (**E**) late childhood ($>$9.0–14.0 years), and (**F**) adolescence ($>$14.0–19.0 years). Models are adjusted for sex of child, and the following maternal characteristics: age at child's birth, education, height, pre-pregnancy BMI, smoking during pregnancy, parity, ethnic background, gestational diabetes and hypertension, and preeclampsia. Cohort-specific estimates were adjusted for the maximum available set of the confounding variables (see Table 1). BMI, body mass index; CI, confidence interval; GA, gestational age; OR, odds ratio.
(TIF)

**S10 Fig. Forest plot of associations between GA (in weeks) and odds of overweight by sex.** Overall adjusted ORs of overweight with 95% CIs from study-specific linear regression models, where cohorts were assigned weights under the random-effects model to attain the OR. The blue (male) and yellow (female) dots in the forest plot represent the adjusted ORs, while the whiskers span the 95% CI, whereas $I^2$-statistics ($I^2$), sample size (N), studies, and $p$-value (two-sided, $<$0.05) relate to adjusted OR. Estimates reflect OR of overweight (vs. normal weight) per week increase in GA at birth in early infancy ($>$0.0–0.5 years), late infancy ($>$0.5–2.0 years), early childhood ($>$2.0–5.0 years), mid-childhood ($>$5.0–9.0 years), late childhood ($>$9.0–14.0 years), and adolescence ($>$14.0–19.0 years). Models are adjusted for sex of child, and the following maternal characteristics: age at child's birth, education, height, prepregnancy BMI, smoking during pregnancy, parity, ethnic background, gestational diabetes and hypertension, and preeclampsia. Cohort-specific estimates were adjusted for the maximum available set of the confounding variables (see Table 1). BMI, body mass index; CI, confidence interval; GA, gestational age; OR, odds ratio.
(TIF)

**S11 Fig. Forest plot of associations between GA (in weeks) and odds of overweight by maternal education.** Overall adjusted ORs of overweight with 95% CIs from study-specific linear regression models, where cohorts were assigned weights under the random-effects model to attain the OR. The blue (high educational level) and yellow (low/medium educational level) dots in the forest plot represents the adjusted OR, while the whiskers span the 95% CI, whereas $I^2$-statistics ($I^2$), sample size (N), studies, and $p$-value (two-sided, $<$0.05) relate to adjusted OR. Estimates reflect OR of overweight (vs. normal weight) per week increase in GA at birth in early infancy ($>$0.0–0.5 years), late infancy ($>$0.5–2.0 years), early childhood ($>$2.0–5.0 years), mid-childhood ($>$5.0–9.0 years), late childhood ($>$9.0–14.0 years), and adolescence ($>$14.0–19.0 years). Models are adjusted for sex of child, and the following maternal characteristics: age at child's birth, education, height, prepregnancy BMI, smoking during pregnancy, parity, ethnic background, gestational diabetes and hypertension, and preeclampsia. Cohort-specific estimates were adjusted for the maximum available set of the confounding variables (see Table 1). BMI, body mass index; CI, confidence interval; GA, gestational age;

OR, odds ratio.
(TIF)

**S12 Fig. Forest plot of associations between GA (in weeks) and odds of overweight by maternal smoking in pregnancy.** Overall adjusted ORs of overweight with 95% CIs from study-specific linear regression models, where cohorts were assigned weights under the random-effects model to attain the OR. The blue (smoking in pregnancy) and yellow (no smoking in pregnancy) dots in the forest plot represents the adjusted OR, while the whiskers span the 95% CI, whereas $I^2$-statistics ($I^2$), sample size (N), studies, and *p*-value (two-sided, <0.05) relate to adjusted OR. Estimates reflect OR of overweight (vs. normal weight) per week increase in GA at birth in early infancy (>0.0–0.5 years), late infancy (>0.5–2.0 years), early childhood (>2.0–5.0 years), mid-childhood (>5.0–9.0 years), late childhood (>9.0–14.0 years), and adolescence (>14.0–19.0 years). Models are adjusted for sex of child, and the following maternal characteristics: age at child's birth, education, height, prepregnancy BMI, smoking during pregnancy, parity, ethnic background, gestational diabetes and hypertension, and preeclampsia. Cohort-specific estimates were adjusted for the maximum available set of the confounding variables (see Table 1). BMI, body mass index; CI, confidence interval; GA, gestational age; OR, odds ratio.
(TIF)

**S13 Fig. Forest plot of associations between GA (in weeks) and odds of overweight by year of birth.** Overall adjusted ORs of overweight with 95% CIs from study-specific linear regression models, where cohorts were assigned weights under the random-effects model to attain the OR. The blue (<2001) and yellow (≥2001) dots in the forest plot represents the adjusted OR, while the whiskers span the 95% CI, whereas $I^2$-statistics ($I^2$), sample size (N), studies, and *p*-value (two-sided, <0.05) relate to adjusted OR. Estimates reflect OR of overweight (vs. normal weight) per week increase in GA at birth in early infancy (>0.0–0.5 years), late infancy (>0.5–2.0 years), early childhood (>2.0–5.0 years), mid-childhood (>5.0–9.0 years), late childhood (>9.0–14.0 years), and adolescence (>14.0–19.0 years). Models are adjusted for sex of child, and the following maternal characteristics: age at child's birth, education, height, prepregnancy BMI, smoking during pregnancy, parity, ethnic background, gestational diabetes and hypertension, and preeclampsia. Cohort-specific estimates were adjusted for the maximum available set of the confounding variables (see Table 1). BMI, body mass index; CI, confidence interval; GA, gestational age; OR, odds ratio.
(TIF)

**S1 Table. Cohort-specific study characteristics and information on exposure and outcome measurements.**
(DOCX)

**S2 Table. Missing values in cohort-specific baseline characteristics.**
(DOCX)

**S3 Table. Results of individual variable meta-regression models showing values of β, se(β), and the significance of β for each study characteristic.**
(DOCX)

**S1 Text. Information about variable classification and coding.**
(DOCX)

**S1 Appendix. Cohort-specific sources of funding/support.**
(DOCX)

**S2 Appendix. Cohort-specific acknowledgments.**
(DOCX)

## Acknowledgments

The authors would like to acknowledge everyone in LifeCycle and EUCAN-Connect who have supported and contributed to each cohort included in the study. In addition, acknowledgments are sent to the DataSHIELD team. Please see S2 Appendix for list of cohort-specific acknowledgments. Also, the authors would like to acknowledge Tanis Fenton for her contribution to this manuscript, and data curation for AOF.

## Author Contributions

**Conceptualization:** Johan L. Vinther, Claus T. Ekstrøm, Thorkild I. A. Sørensen, Deborah A. Lawlor, Anne-Marie Nybo Andersen.

**Data curation:** Johan L. Vinther, Tim Cadman, Demetris Avraam, Claus T. Ekstrøm, Thorkild I. A. Sørensen, Ahmed Elhakeem, Ana C. Santos, Angela Pinot de Moira, Barbara Heude, Carmen Iñiguez, Costanza Pizzi, Elinor Simons, Ellis Voerman, Eva Corpeleijn, Faryal Zariouh, Gilian Santorelli, Hazel M. Inskip, Henrique Barros, Jennie Carson, Jennifer R. Harris, Johanna L. Nader, Justiina Ronkainen, Katrine Strandberg-Larsen, Loreto Santa-Marina, Lucinda Calas, Luise Cederkvist, Maja Popovic, Marie-Aline Charles, Marieke Welten, Martine Vrijheid, Meghan Azad, Padmaja Subbarao, Paul Burton, Puishkumar J. Mandhane, Rae-Chi Huang, Rebecca C. Wilson, Sido Haakma, Sílvia Fernández-Barrés, Stuart Turvey, Susana Santos, Suzanne C. Tough, Sylvain Sebert, Theo J. Moraes, Theodosia Salika, Vincent W. V. Jaddoe, Deborah A. Lawlor, Anne-Marie Nybo Andersen.

**Formal analysis:** Johan L. Vinther, Tim Cadman, Demetris Avraam.

**Funding acquisition:** Hazel M. Inskip, Jennifer R. Harris, Marie-Aline Charles, Martine Vrijheid, Rae-Chi Huang, Vincent W. V. Jaddoe, Deborah A. Lawlor.

**Methodology:** Johan L. Vinther, Tim Cadman, Demetris Avraam, Thorkild I. A. Sørensen, Deborah A. Lawlor, Anne-Marie Nybo Andersen.

**Project administration:** Johan L. Vinther.

**Software:** Johan L. Vinther, Tim Cadman, Demetris Avraam, Paul Burton, Sido Haakma.

**Supervision:** Claus T. Ekstrøm, Deborah A. Lawlor, Anne-Marie Nybo Andersen.

**Visualization:** Johan L. Vinther.

**Writing – original draft:** Johan L. Vinther, Tim Cadman, Demetris Avraam, Claus T. Ekstrøm, Thorkild I. A. Sørensen, Carmen Iñiguez, Deborah A. Lawlor, Anne-Marie Nybo Andersen.

**Writing – review & editing:** Johan L. Vinther, Tim Cadman, Demetris Avraam, Claus T. Ekstrøm, Thorkild I. A. Sørensen, Ahmed Elhakeem, Ana C. Santos, Angela Pinot de Moira, Barbara Heude, Carmen Iñiguez, Costanza Pizzi, Elinor Simons, Ellis Voerman, Eva Corpeleijn, Faryal Zariouh, Gilian Santorelli, Hazel M. Inskip, Henrique Barros, Jennie Carson, Jennifer R. Harris, Johanna L. Nader, Justiina Ronkainen, Katrine Strandberg-Larsen, Loreto Santa-Marina, Lucinda Calas, Luise Cederkvist, Maja Popovic, Marie-Aline Charles, Marieke Welten, Martine Vrijheid, Meghan Azad, Padmaja Subbarao, Paul Burton, Puishkumar J. Mandhane, Rae-Chi Huang, Rebecca C. Wilson, Sido Haakma, Sílvia Fernández-Barrés, Stuart Turvey, Susana Santos, Suzanne C. Tough, Sylvain Sebert,

Theo J. Moraes, Theodosia Salika, Vincent W. V. Jaddoe, Deborah A. Lawlor, Anne-Marie Nybo Andersen.

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
