## [Editor Report · Decision Letter 0]

27 May 2022

Dear Dr Lerbech Vinther, 

Thank you for submitting your manuscript entitled "Gestational age at birth and body size from infancy through adolescence: findings from analyses of individual data on 253,810 singletons in 16 birth cohort studies" for consideration by PLOS Medicine.

Your manuscript has now been evaluated by the PLOS Medicine editorial staff and I am writing to let you know that we would like to send your submission out for external peer review.

Please re-submit your manuscript within two working days, i.e. by May 31 2022 11:59PM.

Kind regards,

Caitlin Moyer, Ph.D.

Associate Editor

PLOS Medicine

---

## [Decision Letter · Decision Letter 1]

9 Nov 2022

Dear Dr. Lerbech Vinther,

Thank you very much for submitting your manuscript "Gestational age at birth and body size from infancy through adolescence: findings from analyses of individual data on 253,810 singletons in 16 birth cohort studies" (PMEDICINE-D-22-01655R1) for consideration at PLOS Medicine. 

[LINK]

In light of these reviews, I am afraid that we will not be able to accept the manuscript for publication in the journal in its current form, but we would like to consider a revised version that addresses the reviewers' and editors' comments. Obviously we cannot make any decision about publication until we have seen the revised manuscript and your response, and we plan to seek re-review by one or more of the reviewers. 

We expect to receive your revised manuscript by Nov 30 2022 11:59PM. Please email us (plosmedicine@plos.org) if you have any questions or concerns.

We look forward to receiving your revised manuscript. 

Sincerely,

Philippa Dodd, MBBS MRCP PhD

PLOS Medicine

plosmedicine.org

Please address all editor and reviewer comments below in full

COMMENTS FROM THE ACADEMIC EDITOR

I agree to the authors statement that they should not adjust for birth weight, since this is on the causal pathway.

But maybe they should comment on why they don't analyze whether the associations partly are explained by small-for gestational age at birth, given available evidence within the DOHaD paradigm on intrauterine growth restriction and later nutritional status and health?

The included studies all represent high-income settings. It would be valuable to get comments on the limitation that low-or middle-income settings with a rapid nutritional transition are not represented, and what potential consequences this may have for the findings.

GENERAL

Please report your IPD MA according to the PRISMA guidelines provided at the EQUATOR site. http://www.equator-network.org/reporting-guidelines/prisma/

Please provide the completed PRISMA checklist. When completing the checklist, please use section and paragraph numbers, rather than page numbers. Please add the following statement, or similar, to the Methods: "This study is reported as per the Preferred Reporting Items for Systematic Reviews and Meta-Analyses (PRISMA) guideline (S1 Checklist)."

DATA AVAILABILITY STATEMENT

Please revise your data availability statement. Referring to another published manuscript regarding data access will not suffice.

PLOS Medicine requires that the de-identified data underlying the specific results in a published article be made available, without restrictions on access, in a public repository or as Supporting Information at the time of article publication, provided it is legal and ethical to do so. Please see the policy at http://journals.plos.org/plosmedicine/s/data-availability and FAQs at http://journals.plos.org/plosmedicine/s/data-availability#loc-faqs-for-data-policy

For each data source used in your study: 

TITLE

Please revise your title according to PLOS Medicine’s style. Your title must be nondeclarative and not a question. It should begin with main concept if possible. "Effect of" should be used only if causality can be inferred, i.e., for an RCT. Please place the study design ("A randomized controlled trial," "A retrospective study," "A modelling study," etc.) in the subtitle (ie, after a colon).

ABSTRACT

Please report your abstract according to PRISMA for abstracts, following the PLOS Medicine abstract structure (Background, Methods and Findings, Conclusions) http://www.plosmedicine.org/article/info:doi/10.1371/journal.pmed.1001419

Abstract methods and findings:

Please provide details of the process of selection for the 16 population cohort studies included in your study - as you have done in main methods section

Please provide data sources and eligibility criteria. 

Please provide dates between which the 16 included studies were published 

Please include the important dependent variables that are adjusted for in the analyses

Please quantify the main results p-values as well as with 95% CIs 

Please define GA at first use

Please define OR and 95% CI at first use

In the last sentence of the Abstract Methods and Findings section, please describe the main limitation(s) of the study's methodology.

Abstract conclusions:

Please emphasize what is new without overstating your conclusions

AUTHOR SUMMARY

METHODS and RESULTS

We ask the following of all authors who submit a SR/MA, some may not apply to your IPD MA. Please respond in full to the below points to help facilitate and maximize transparency of data reporting. In any instance(s) where you are unable to respond to the request, please clearly state the reason(s) why not: 

• Please update your included studies to the present time

• Please evaluate study quality and risk of bias

• Please evaluate evidence of publication bias

Please provide dates between which the 16 included studies were published (as for the abstract).

Please provide the actual numbers of events for the outcomes, not just summary statistics or ORs.

Please present numerators and denominators for percentages

Where relevant, where you report 95% CIs, please also report p-values, including in the tables of the main manuscript and supplementary files. When a p-value is given please also provide the statistical test used to determine it. Please specify the significance level used (eg, P<0.05, two-sided). Please do not report p=0.000; report as p < 0.001

How was “maternal ethnicity” defined and by whom? Why was ethnicity considered important in this study and what it is believed to represent?

Please update your ethics statement and include details of individual study ethical approval as a paragraph and not a list, in the main manuscript and not in the supplementary files. Please include in each case whether consent was written or oral.

FIGURES & TABLES

Where you present adjusted analyses, please also present unadjusted analyses including in the supplementary figures. 

Please also include p-values where you report 95% CIs including supplementary figures

Please indicate, in each figure captions what the dot and whiskers represent in the forest plots

Please ensure that all abbreviations e.g. BMI and CI are defined within each figure caption in the main manuscript figures and supplementary figures.

Table 3: in the figure caption, please define BMI

Table S4: please define β and β (se) in the caption

Please ensure that any colours you use in the figures are accessible to those with colour blindness (i.e. avoid the use of green and/or red)

DISCUSSION

Please remove sub-headings from the discussion, including that which depicts the conclusions, such that the discussion reads as one continuous piece of prose.

Please remove the funding statement and conflict of interest statement form the end of the main manuscript and include only in the submission form. 

REFERENCES

Please see our website for reference guidelines here: https://journals.plos.org/plosmedicine/s/submission-guidelines#loc-references

Citations should be placed within square parentheses preceding punctuation. Where more than one article is cited please separate the citations with a comma and no spaces as follows: [1,2,3,4] or [1-3,5,7]

Journal name abbreviations should be those found in the National Center for Biotechnology Information (NCBI) databases.

In the reference list please ensure no more than 6 author names are listed followed by et al. where more than 6 co-authors contribute to the study.

Comments from the reviewers:

Reviewer #1: The authors investigated the association of GA with BMI z-score and odd of overweight from infancy through adolescence. They found interesting results that higher GA is important for higher BMI in infancy, while the association attenuate with age and preterm children have a similar mean BMI to those born term.

Major points

・I could not understand how you calculated BMI z-score during infancy as in references (68) and (69) because there was no BMI z-score data for under five years. 

・I think you should add the data of small for gestational age at birth and analyze the influence of SGA.

Reviewer #2: This is a well-conducted study on the association between gestational age at birth and body size from infancy to adolescence in 16 birth cohort studies. The study design, datasets, statistical methods and analyses, and presentation (tables and figures) and interpretation of the results are mostly adequate. However, there are still a few major issues needing attention.

1) Year of birth. From Table 1 we can see the year of birth ranges from 1985 to 2017 - across 33 years for the 16 cohort studies. This big time difference needs to be addressed in the analyses as the landscape of perinatal mortality, morbidity and healthcare has changed a lot over time. However, year of birth was not adjusted at all in any of the analyses in the paper which is inadequate. Firstly, year of birth need be adjusted in all the analyses; secondly, analyses such as meta regression on whether this is a time trend of the results are needed.

2) Interpretation of results for adolescence. One of the key conclusions in the abstract is "By adolescence, preterm children have on average a similar mean BMI to those born term". However, as shown in figure 2 (also figure 1), there are only 4 studies included in the analysis for adolescent outcomes as compared to 16 or so studies for analyses of earlier years' outcomes, therefore the sample size is substantially reduced for the analyses for this adolescent category, which may impact on the statistical significance of the results. This issue happens to all the analyses for the adolescence in the paper. For example, in figure 2, the no difference in mean BMI between preterm and term for adolescence may be due to the very wide 95% CIs caused by smaller sample size so that the results might be inconclusive due to uncertainty. Overall, this is a limitation of the study and the results need to be read with caution.

3) Missing data. The missing percentages are shown in S3 Table and we can see some studies have high missing rates in some variables. It looks like the authors have used available data for analyses. Have the authors dealt with the missing data in any way? If not, what are the impact and limitaion to the results caused by missing data potentially?

4) 'Leave one out' analysis is a good option for sensitivity analysis. The results are shown in S2 figure but a bit difficult to follow. For example, in S2 figure A, what's estimate for each cohort study? Does it mean leave that named study out? Basically, I'd like to see the results without the BIB study from the UK as it shows very different characteristics and outcomes (and also missing data) from the others as shown in table 1 and S3.

Reviewer #3: This is an important study using data from multiple cohorts addressing the association between gestational age and later BMI. The paper is well-written, and the methodology is fine.

I have several comments:

1. The ethical paragraph is too brief. In fact, the authors only refer to the supplements for a description about the MREC/IRB approval and the consent procedure of each study. This is insufficient. The ethical paragraph should describe the grounds for allowance of data sharing with an international consortium. More specifically, did parents give written informed consent for data sharing, or was a waiver of consent for data sharing provided (by the local MREC/IRB or the local privacy officer) under the assumption that data could impossibly identify individual subjects (eg, due to the large sample size, sharing of minimal data)? In this regard, it is also important to clarify who was the receiving party.

2. I would be more careful with the conclusion that "by adolescence, preterm children have on average a similar mean BMI to those born full term". There is no follow-up beyond 19 years, implicating that the observed catch-up in BMI after prematurity, as observed at age categories 9-14 and 14-19, might be determined by earlier pubertal timing. There is some evidence suggesting that children born preterm have an earlier onset of pubertal growth velocity (Wehkalampi, et all. JCEM 2011), although Tanner pubertal stages and menarcheal age are generally age-appropriate (Ford, et al. Arch Pediatr Adolesc Med 2000; Peralta-Carcelen, et al. J Pediatr 2001; Hack, et al. Pediatrics 2003; Saigal, et al. Pediatr Res 2006). The authors should add to the Discussion that pubertal timing might play a role in the associations observed.

3. Limitations are generally well acknowledged, but in view of my previous comment some issues should be added to this section, including (1) the lack of follow-up extending into adulthood, (2) the low proportion of preterm born children, and (3) the lack of information on puberty status (at least for the age categories 9-14 and 14-19).

4. I am pleased to read that the authors clearly state in the Abstract and in the Introduction that birth weight cannot be used as a proxy for gestational age, a mistake that has been made by many previous studies, eg, due to lack of accurate gestational age determination in older cohorts. It would be nice if the authors could place their findings into (historical) context in the Discussion.

Reviewer #4: Summary and Recommendation: 

1) This study provides a novel approach to analyzing a unique dataset from the EU Child Cohort Network to investigate the relationship of gestational age/prematurity to later body size from infancy through adolescence. By including individual data on 253,810 children from 16 cohorts across a wide range of gestational ages and follow-up ages and by controlling for covariates and assessing heterogeneity, this study aims to overcome a number of methodological limitations of prior published studies, reviews and meta-analyses. The result is a robust evaluation of a large and diverse population, though it is still limited by a number of factors, including those which the authors point out as well as a number of other issues that need to be further addressed.

2) The authors employ a linear regression model to exam the associations of GA in weeks and in clinical categories (5 groups ranging from very preterm to post-term) with BMI z-scores across 6 age bands (ranging from early infancy to adolescence); and they employ a binomial logistic regression model to examine the relationship of GA and of clinical categories to overweight (defined as >2 SD above WHO Child Growth Standard median for children < 5yo and >1 SD above WHO Growth Reference median for children >=5yo) across similar age bands. Both sets of analyses control for multiple a priori selected factors that were known or plausible causes of variation in GA and subsequent body size, with final selection based on a directed acyclic graph. For 8 cohorts where multiple growth measurements were available for the same child in one or more of the six age-groups the latest measurement was chosen, presumably in order to treat last age of measurement as a cross-sectional variable across all cohorts. 

3) There were multiple differences in cohort-specific sample size, distributions of covariates, gestational age groups and follow-up age groups. The proportion of children classified as overweight also varied between cohorts in each age group.

4) Given this heterogeneity, the authors performed meta-regression analyses to determine which study characteristics were independently associated with between-study (cohort) heterogeneity and also performed a 'leave-one-out' analysis for cross-validation.

5) Would recommend revisions to address issues noted below

Evidence and Examples

1) Major Issues: 

a. In addition to the heterogeneity that was analyzed, there was also variation in sources of growth data as delineated in Table S1, including at least 4 studies where the source was parent report, including the two largest cohorts from Denmark and Norway (which together comprise over 2/3 of the total sample). A further analysis to compare data from the studies with actual measurements with those from studies with parent-reported measurements would be important. 

b. There was also heterogeneity in years of birth (ranging from 1985 to 2017 and presumably also in years at last follow-up measurement. As the authors suggest in the discussion this may reflect some changes over time in population and newborn and subsequent management; but no analysis by year of birth is provided. 

c. The percent of children born preterm ranged from 3.1% to 7.5%, which is relatively low compared with other reports (even excluding extremely preterm infants), with 4 cohorts including <1% very preterm. 

d. Granted following children up through adolescence is challenging, the body size data for >14 - 19 yo was based on only 4 studies, with over ¾ of those coming from one cohort (Denmark). This needs to be mentioned in limitations section. 

e. The results for BMI scores indicate that increasing GA is associated with increasing BMI only during later infancy/early childhood, with progressive attenuation up through adolescence. Conversely, the more premature the clinical categories the lower the BMIz compared to term (for very preterm this was observed at all ages up until adolescence and for late until pre-adolescence). At first glance this appears paradoxical, but on further consideration it is likely related to the different "directions" of the comparison (higher GA compared with lower in the first case and lower compared with higher (term) in the second. That said, this would be worth clarifying further in the discussion. 

f. A similar relationship of GA to overweight was found, with attenuation through childhood to adolescence; but in contrast to BMI results and to overweight results by GA, very preterm infants had an increased risk of overweight in adolescence. Though the latter is consistent with several other cited studies from Finland and Australia, the reason for this shift at adolescence is not clear from the data presented and though the discussion points this out, it does not elucidate further. Several recent studies of extremely preterm infants found high prevalence of obesity at pre-adolescence or adolescence, but no difference in prevalence compared to term controls (a study from EPICURE network in England Yanyan Ni et al Arch Dis Child Fetal Neonatal Ed. 2020 Sep;105(5):496-503 and a study from Dallas Jessica Wickland et al Pediatr Res April 2022); yet other co-morbidities were found elevated. This would be worth a brief comment. 

2) Minor Issues:

a. Results note several associations in linear regression analyses whose confidence intervals include 0; strictly speaking such associations are not statistically significant, even though the estimated mean difference in BMIz is >0.0. Moreover, the Forest Plots for these results in Figure 1 do not clearly show inclusion of x axis =0 in the plots. 

b. Likewise, in the results from the logistic regression, several associations are noted which include 1 in the confidence interval; strictly speaking such associations are not statistically significant, even though the odds ratio is>1.0.

[LINK]

---

## [Decision Letter · Decision Letter 2]

12 Dec 2022

Dear Dr. Lerbech Vinther,

Thank you very much for re-submitting your manuscript "Gestational age at birth and body size from infancy through adolescence: findings from analyses of individual data on 253,810 singletons in 16 birth cohort studies" (PMEDICINE-D-22-01655R2) for review by PLOS Medicine.

I have discussed the paper with my colleagues and the academic editor and it was also seen again by 3 reviewers. I am pleased to say that provided the remaining editorial and production issues are dealt with we are planning to accept the paper for publication in the journal.

[LINK]

We look forward to receiving the revised manuscript by Dec 19 2022 11:59PM.   

Sincerely,

Philippa Dodd, MBBS MRCP PhD

PLOS Medicine

plosmedicine.org

Requests from Editors:

GENERAL

Thank you for your considered and detailed response to previous editor and reviewer requests, all of which are satisfactory upon review. 

The editorial team notes that upon comparison of your manuscript versions many, of the required changes, despite being detailed clearly in your responses were not evident in the manuscript – for example, the author summary is still absent, some figures appear unchanged, changes to the abstract are not apparent.

*** Please see below for specific examples and please check carefully against ALL previous requests to ensure that all changes have been included in the manuscript ***

TITLE

We note your response to the editorial request to update your title. Please revise your title such that it complies with our journal requirements. It should be defined as an individual participant data meta-analysis. Suggest the following:

“Gestational age at birth and body size from infancy through adolescence: an individual participant data meta-analysis of 253,810 singletons from 16 birth cohort studies”

ABSTRACT

We requested that p-values be added to the data reporting, which is a journal requirement, as you acknowledged in your response. We could not see any p-values in the abstract. Please add the p-values. Please also ensure that CIs are reported consistently, currently upper and lower limits are separated either by a semi-colon or a colon and should be separated by a comma (or hyphen although this can cause confusion with negative value reporting) please revise accordingly and throughout the main manuscript. 

We requested that the main limitations of the study’s methodology be detailed at the end of the methods and findings, but we could see it. Please include this in the abstract.

AUTHOR SUMMARY

We could not see an author summary in the main manuscript, please include as previously requested following the abstract

METHODS and RESULTS

Many of the figures are unchanged compared to previous versions – for example unadjusted analyses are not present, p-values are absent from the main manuscript

REFERENCES

We previously asked:

“Please see our website for reference guidelines here: https://journals.plos.org/plosmedicine/s/submission-guidelines#loc-references

Citations should be placed within square parentheses preceding punctuation. Where more than one article is cited please separate the citations with a comma and no spaces as follows: [1,2,3,4] or [1-3,5,7]

Journal name abbreviations should be those found in the National Center for Biotechnology Information (NCBI) databases.

In the reference list please ensure no more than 6 author names are listed followed by et al. where more than 6 co-authors contribute to the study.”

These changes were not evident, please amend as previously requested

SOCIAL MEDIA

To help us extend the reach of your research, please provide any Twitter handle(s) that would be appropriate to tag, including your own, your coauthors’, your institution, funder, or lab. Please respond to this email with any handles you wish to be included when we tweet this paper.

Comments from Reviewers:

Reviewer #2: Thanks authors for their great effort to improve the manuscript. I am satisfied with the response and revision. No further issues needing attention.

Reviewer #3: The paper is acceptable for publication.

Reviewer #4: The authors have addressed the issues raised by the reviewers to reasonable degree, and included in discussion those issues it was not feasible to analyze further.

[LINK]

---

## [Editor Report · Decision Letter 3]

19 Dec 2022

Dear Dr Lerbech Vinther, 

On behalf of my colleagues and the Academic Editor, Professor Lars Persson, I am pleased to inform you that we have agreed to publish your manuscript "Gestational age at birth and body size from infancy through adolescence: an individual participant data meta-analysis on 253,810 singletons in 16 birth cohort studies" (PMEDICINE-D-22-01655R3) in PLOS Medicine.

We require the following be addressed prior to publication:

AUTHOR SUMMARY - please split your author summary into individually bulleted points. We suggest that each sentence becomes an individual point under the relevant sub-heading. Please note that no more than 4 bulleted points should be listed under each sub-heading.

PRESS

Best wishes, 

Philippa Dodd, MBBS MRCP PhD 

PLOS Medicine